# SAVE: A Generalizable Framework for Multi-Condition Single-Cell Generation with Gene Block Attention

**Jiahao Li    Jiayi Dong    Peng Ye    Xiaochi Zhou    Haohai Lu    Fei Wang***
**College of Computer Science and Artificial Intelligence, Fudan University**
**Shanghai Key Laboratory of Intelligent Information Processing, Fudan University**
`{lijiahao23, jiayidong21, pye24, xczhou24,`
`hhlu24}@m.fudan.edu.cn, wangfei@fudan.edu.cn`

## Abstract

Modeling single-cell gene expression across diverse biological and technical conditions is crucial for characterizing cellular states and simulating unseen scenarios. Existing methods often treat genes as independent tokens, overlooking their high-level biological relationships and leading to poor performance. We introduce SAVE, a unified generative framework based on conditional Transformers for multi-condition single-cell modeling. SAVE leverages a coarse-grained representation by grouping semantically related genes into blocks, capturing higher-order dependencies among gene modules. A Flow Matching mechanism and condition-masking strategy further enhance flexible simulation and enable generalization to unseen condition combinations. We evaluate SAVE on a range of benchmarks, including conditional generation, batch effect correction, and perturbation prediction. SAVE consistently outperforms state-of-the-art methods in generation fidelity and extrapolative generalization, especially in low-resource or combinatorially held-out settings. Overall, SAVE offers a scalable and generalizable solution for modeling complex single-cell data, with broad utility in virtual cell synthesis and biological interpretation. Our code is publicly available at `https://github.com/fdu-wangfeilab/sc-save`

## 1 Introduction

The rapid expansion of high-throughput single-cell RNA sequencing (scRNA-seq) technologies provides unprecedented opportunities to computationally model and simulate diverse cellular states under complex experimental conditions Kolodziejczyk et al. (2015); Svensson et al. (2018); Jovic et al. (2022). Generative models that can predict gene expression profiles under unseen combinations of covariates—such as cell type, disease state, and perturbation—can greatly reduce experimental costs and accelerate biological discovery.

Variational Autoencoders (VAEs) have become the foundation of many generative frameworks in single-cell omics, with scVI Lopez et al. (2018) being a notable example. While effective in learning latent representations and enabling batch correction and imputation, such models typically assume simple architectures and are limited in their ability to model complex, combinatorial interactions among multiple external conditions. Addressing this limitation is crucial for simulating realistic cellular responses in multi-conditional settings Luecken et al. (2024).

To model condition-dependent gene expression patterns at scale, recent approaches have adopted Transformer-based architectures, representing cells as gene token sequences and conditions as condition tokens Cui et al. (2024); Bian et al. (2024). These so-called "foundation models" rely on masked modeling objectives to capture the relationships between genes and covariates. However, they face several fundamental challenges: (1) they assume a flat, token-level view of genes, neglecting biological structures such as gene modules or pathways; (2) they fail to model the global expression distribution, focusing primarily on non-zero values and ignoring the informative zero inflation inherent

---

*Corresponding author.

in scRNA-seq data Qiu (2020); and (3) they are typically not integrated into a generative framework capable of sampling from the learned conditional distribution.

We draw inspiration from masked generative modeling in computer vision, such as Chang et al. (2022), which demonstrates that unordered data can be effectively modeled using coarse-grained representations like image patches. Gene expression data shares key properties with images in this context—it is high-dimensional, sparse, and inherently unordered. This suggests that a similar coarse-graining strategy, modeling at the level of 'gene blocks' rather than individual genes, could prove highly effective.

However, a key challenge arises: unlike pixels, which are grouped by spatial proximity, genes lack a natural local structure. To overcome this, we propose forming blocks based on semantic similarity. We leverage Large Language Models (LLMs), pre-trained on extensive text corpora, to extract rich features from the comprehensive gene descriptions in the NCBI database. Genes exhibiting high semantic similarity are then aggregated into blocks. These blocks serve as coarser-grained tokens, upon which we apply an attention mechanism to learn the complex relationships among them.

In this work, we propose SAVE (**S**ingle-cell Gene Block **A**ttention-based **V**ariational g**E**nerative framework), a unified framework for conditional single-cell data generation and integration. SAVE integrates the latent space structure of VAEs with a coarse-grained Transformer that attends over meaningful gene blocks, effectively capturing high-order dependencies. To enhance conditional generation, SAVE employs a Flow Matching , enabling simulation under complex, unseen condition combinations.

Our main contributions are summarized as follows:

- We introduce Gene Block Attention, a attention mechanism that captures high-order relationships among blocks of genes.
- We develop masked modeling strategy on Flow Matching and VAE to enhance SAVE's ability to learn the conditional distribution of cell states.
- We demonstrate that SAVE achieves state-of-the-art performance on multiple tasks, including batch alignment, perturbation prediction, and simulation under unseen condition combinations.

## 2 RELATED WORK

### 2.1 GENERATIVE MODELING FOR SINGLE-CELL DATA

Variational Autoencoders (VAEs) have been widely adopted for modeling single-cell RNA-seq data. Notably, scVI Lopez et al. (2018) introduces a zero-inflated negative binomial (ZINB) likelihood and encodes covariates into the latent space for tasks like batch correction and imputation. However, traditional VAE-based models primarily focus on representation learning rather than flexible data generation across complex conditions Xiong et al. (2022). To improve generative capabilities, recent models have explored more expressive architectures. CFGen Palma et al., a flow-based model, applies optimal transport-guided diffusion and classifier-free guidance to model conditional distributions. Similarly, scDiffusion Luo et al. (2024) combines latent diffusion with a pre-trained autoencoder Heimberg et al. (2025) and integrates condition labels via gradient-based classifier guidance. While these approaches support conditional generation, they rely on fine-grained diffusion processes and often struggle with interpretability and scalability across diverse biological contexts.

### 2.2 CONDITIONAL TRANSFER IN SINGLE-CELL ANALYSIS

Predicting gene expression under unseen conditions—such as novel drug perturbations or disease states—is a central goal in single-cell analysis Wu et al. (2024). scGen Lotfollahi et al. (2019) addresses this using a latent shift strategy in an autoencoder framework, assuming linear transitions in the latent space. trVAE Lotfollahi et al. (2020) applies Maximum Mean Discrepancy (MMD) for alignment and decodes data conditioned on state labels. These methods, however, are typically limited to a single type of condition. scDisInFact Zhang et al. (2024) extends to multi-factor settings by employing multiple encoders to disentangle condition-specific and condition-invariant

components. Nonetheless, such designs require predefined factor separation and often lack scalability to combinatorial condition spaces.

## 2.3 TRANSFORMER MODELS IN SINGLE-CELL LEARNING

Transformer-based models have recently gained traction in single-cell omics, mostly for representation learning. scGPT Cui et al. (2024), scBERT Yang et al. (2022), and Geneformer Theodoris et al. (2023) tokenize gene expression using rank or discretized values, often sacrificing fine-grained quantitative information. GeneCompass Yang et al. (2024) improves upon this by combining rank and absolute expression with regression loss. Some methods directly project expression values into continuous space, with additional strategies to handle zero inflation. TOSICA Chen et al. (2023) uses gene networks to filter unreliable zeros, while scFoundation Hao et al. (2024) introduces special tokens to mask them. CellPLM Wen et al. (2023) models inter-cell relationships via a VAE-Transformer hybrid, treating entire cells as tokens. Despite this progress, most Transformer-based methods focus on encoding rather than generation, and there is little agreement on how best to represent gene expression in tokenized form. Incorporating biological structure (e.g., gene sets) and enabling conditional generation remain open challenges.

## 3 METHODOLOGY

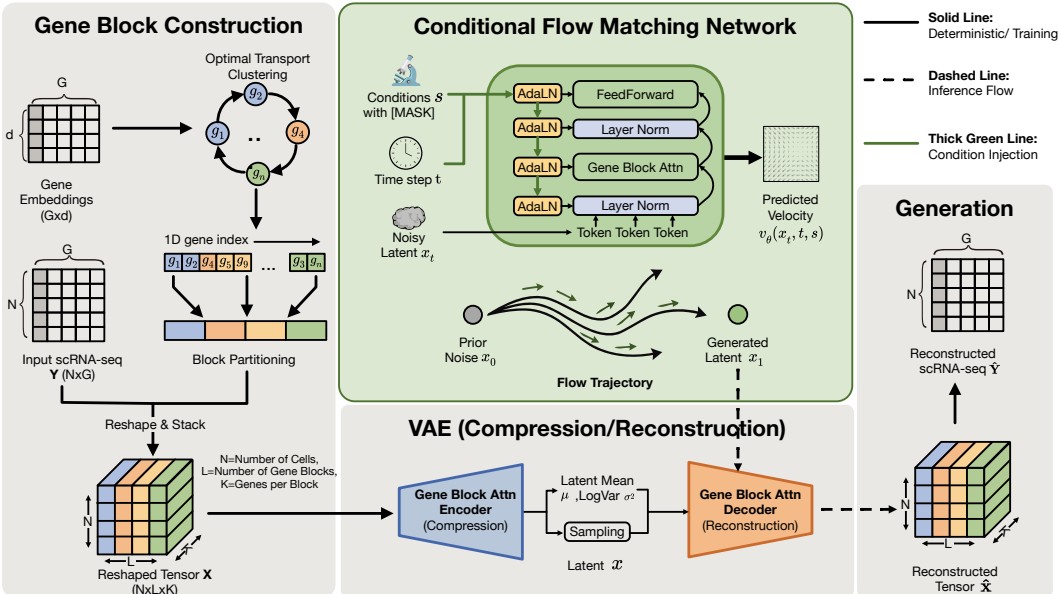

Figure 1: **The overall architecture of the SAVE framework.** The model consists of three main modules: (1) **Gene Block Construction**, which partitions gene embeddings into semantic blocks via Optimal Transport clustering; (2)a **VAE** utilizing Gene Block Attention for latent compression and reconstruction ; and (3) a **Conditional Flow Matching Network**, which maps prior noise $x_0$ to generated latent distributions $x_1$ by integrating conditions $s$ and time steps $t$ via Adaptive Layer Normalization (AdaLN).

We present SAVE, a unified Latent Flow Matching (LFM) framework for conditional simulation and status transformation of scRNA-seq data. To model complex gene expression profiles, SAVE introduces a Gene Block Attention backbone to capture long-range transcriptional dependencies. Furthermore, it incorporates rich contextual information, such as cell type or disease state, through Adaptive Layer Normalization (AdaLN). The architecture comprises three core components: (1) A VAE Encoder with Gene Block Attention to learn robust latent cell representations from transcriptional patterns. (2) A Condition-aware Flow Matching module that leverages AdaLN to generate gene expression profiles precisely guided by condition embeddings. (3) A Condition Mask-based Training

strategy that unifies generation and transfer tasks by masking conditions for either the Flow Matching module or the VAE, respectively. An overview is shown in Figure 1.

## 3.1 GENE BLOCK ATTENTION FOR SCRNA-SEQ MODELING

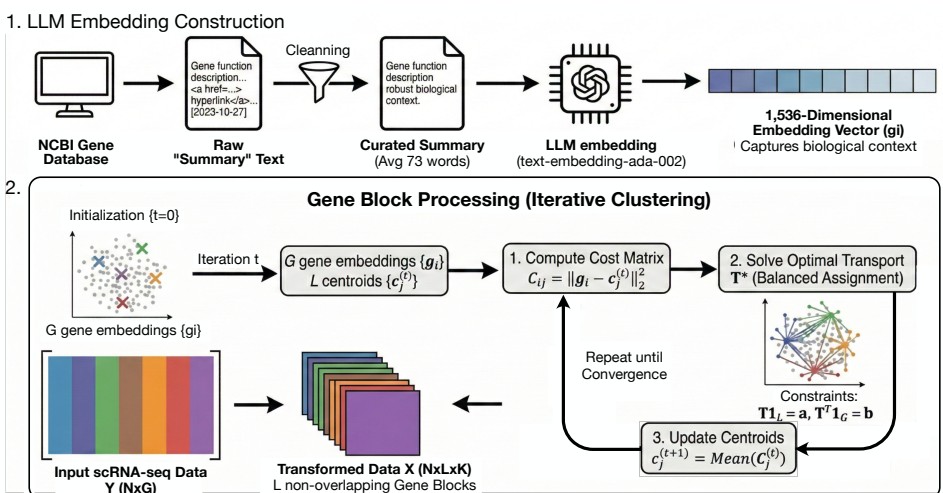

Figure 2: **Overview of the Gene Block Construction process.** The procedure is divided into two modules. **(1) LLM Embedding Construction**: Raw gene function descriptions from the NCBI database are cleaned and processed by LLM to extract semantic embeddings ($g_i$) representing robust biological contexts. **(2) Gene Block Processing**: An iterative clustering algorithm driven by Optimal Transport assigns $G$ gene embeddings into $L$ balanced, non-overlapping blocks.

**Gene Block Processing.** We first introduce our strategy for transforming flat single-cell expression profiles into a coarse-grained, biologically meaningful representation. As illustrated in Figure 2, the construction of Gene Blocks consists of two main stages: Large Language Model (LLM) embedding construction and iterative clustering via Optimal Transport. First, following Chen & Zou (2024), we extract and clean "Summary" texts from the NCBI Gene database to remove uninformative elements. These curated summaries are then processed by an LLM (text-embedding-ada-002) to generate 1,536-dimensional embedding vectors ($g_i$) that capture rich biological context. Second, to group these semantic embeddings into $L$ non-overlapping blocks of equal size, we formulate the block assignment as an Optimal Transport problem. By iteratively computing the Euclidean cost matrix $C_{ij} = ||g_i - c_j^{(t)}||_2^2$, solving for a balanced transport plan with uniform marginal constraints ($T\mathbf{1}_L = a$, $T^T\mathbf{1}_G = b$), and updating the block centroids $c_j^{(t)}$, the algorithm converges to structurally balanced functional modules. This process ultimately reshapes the input scRNA-seq data $Y$ into a structured tensor $X$, serving as the foundational input for our Gene Block Attention mechanism.

**Transformer Block.** Each gene block is first projected into an $e$-dimensional hidden space via a learnable MLP $W^{in}$. SAVE applies standard Transformer blocks to this representation using the following formulation:

$$
\begin{aligned}
h_0 &= XW^{in}, h_0 \in \mathbb{R}^{N \times L \times e} \\
h_{t'} &= h_t + \text{Attention}(\text{LayerNorm}(h_t)) \\
h_{t+1} &= h_{t'} + \text{FeedForward}(\text{LayerNorm}(h_{t'}))
\end{aligned}
\tag{1}
$$

Here, $h_t$ denotes the input to the $t$-th Transformer block. Layer normalization is applied before Attention and FeedForward layers to enhance training stability and convergence.

## 3.2 CONDITION INJECTION VIA ADAPTIVE LAYER NORMALIZATION

To incorporate condition-specific information, we encode all conditioning variables (e.g., batch, cell type, disease stage) into a matrix $S \in \mathbb{R}^{N \times d_s}$, where $d_s$ is the number of condition types. Each

categorical condition is assigned a unique index value, and we apply a learnable embedding to obtain $S^E \in \mathbb{R}^{N \times d_s \times e}$.

We employ Adaptive Layer Normalization (AdaLN) Xu et al. (2019) to inject condition-specific signals into the Transformer blocks. The parameters for AdaLN are derived from $S^E$ as follows:

$$\alpha_1, \beta_1, \gamma_1, \alpha_2, \beta_2, \gamma_2 = S^E W^S, \quad \alpha, \beta, \gamma \in \mathbb{R}^{N \times e} \tag{2}$$

$$h_{t'} = h_t + \alpha_1 \cdot \text{Attention}(\text{AdaLN}(h_t, \gamma_1, \beta_1)) \tag{3}$$

$$h_{t+1} = h_{t'} + \alpha_2 \cdot \text{FeedForward}(\text{AdaLN}(h_{t'}, \gamma_2, \beta_2)) \tag{4}$$

$$\text{AdaLN}(h, \gamma, \beta) = \frac{h - E[h]}{\sqrt{\text{Var}[h] + \epsilon}} \cdot \gamma + \beta \tag{5}$$

Here, $\alpha, \beta, \gamma$ act as learnable scaling factors, modulating the influence of condition embeddings on the sequence representation.

### 3.3 MASK MODELING STRATEGY

To enhance the model's ability to generalize to unseen conditions, we introduce a masking strategy. Each element in the condition matrix $S$ is masked independently with a fixed probability $p$, replaced by a dedicated `[MASK]` token:

$$S_{ij} = \begin{cases} \texttt{[MASK]} & \text{with probability } p \\ S_{ij} & \text{with probability } 1 - p \end{cases} \quad \forall i \in \{1, \ldots, N\}, j \in \{1, \ldots, d_s\} \tag{6}$$

This masking strategy helps the model learn robust representations by simulating missing information during flow matching.

### 3.4 COMPRESSION AND RECONSTRUCTION VIA VARIATIONAL AUTOENCODER

**Attention Encoder with Gaussian Prior.** The encoder's final output $h_i$ is flattened and projected to estimate the parameters of the latent distribution:

$$\mu = (h)W^\mu, \quad \mu \in \mathbb{R}^{N \times L \times d} \tag{7}$$

$$\sigma^2 = (h)W^\sigma, \quad \sigma^2 \in \mathbb{R}^{N \times L \times d} \tag{8}$$

To regularize the latent space, we apply a Kullback–Leibler divergence penalty:

$$\mathcal{L}_p = D_{KL}(\mathcal{N}(\mu, \sigma^2) \| \mathcal{N}(0, 1)) = \frac{1}{2} \left( -\log(\sigma^2) + \sigma^2 + \mu^2 - 1 \right) \tag{9}$$

Latent codes $x$ are sampled using the reparameterization trick.

**Attention Decoder for Latent Tokens.** The decoder mirrors the encoder structure. We apply a linear projection $W^{Din}$ to obtain $h_0^D$, the initial hidden state:

$$h_0^D = xW^{Din}, h_0^D \in \mathbb{R}^{N \times L \times e} \tag{10}$$

$$h = Decoder(h_0^D), \ \hat{X} = hW^{out} \tag{11}$$

The reconstruction loss is defined as: $\mathcal{L}_{recon} = -\log L(\hat{X}|X)$

### 3.5 FLOW MATCHING FOR CONDITIONAL GENERATION

The core idea of Flow Matching is to learn a time-dependent vector field $v_t(x)$ that generates a probability path $p_t(x)$ connecting a simple prior distribution $p_0$ (e.g., a standard Gaussian) to the data distribution $p_1$. The generation process is then described by the probability flow ODE: $\frac{dx_t}{dt} = v_t(x_t)$.

Instead of learning the complex marginal vector field $v_t$ directly, Flow Matching regresses a simpler conditional vector field $u_t$ that maps a specific noise sample $x_0 \sim p_0$ to a specific data sample $x_1 \sim p_1$. This is achieved by defining a probability path between them.

We utilizes a simple yet effective Affine Probability Path, which corresponds to a linear interpolation between the noise and the data. Given a random time step $t \in [0, 1]$, a point $x_t$ on the path is defined as:

$$x_t = (1 - t)x_0 + tx_1 \tag{12}$$

This formulation defines where a particle starting at $x_0$ should be at time t to reach $x_1$ at time $t = 1$.

The training objective is to teach a neural network, $v_\theta(x, t, s)$, to predict the instantaneous velocity of the particle along the path. For the affine path, this velocity, or the target vector field $u_t$, is simply the time derivative of $x_t$ :

$$u_t = \frac{dx_t}{dt} = \frac{d}{dt}((1 - t)x_0 + tx_1) = x_1 - x_0 \tag{13}$$

The network $v_\theta$ takes the perturbed data $x_t$, the timestep t, and optional conditioning information c as input. It is trained to approximate $u_t$ by minimizing the Flow Matching objective, which is a Mean Squared Error (MSE) loss between the predicted vector and the ground-truth vector. The loss function is formulated as:

$$\mathcal{L}_{FM}(\theta) = \mathbb{E}_{t \sim U[0,1], p_0(x_0), p_1(x_1)} \left[ \|v_\theta(x_t, t, s) - u_t\|^2 \right] \tag{14}$$

Once the model $v_\theta$ is trained, it can generate new samples. The generation process reverses the flow, starting from a random noise sample and evolving it toward the data distribution. This is achieved by solving the probability flow ODE from $t = 0$ to $t = 1$, using the learned network $v_\theta$ as the velocity field function. The ODE is defined as:

$$\frac{dx_t}{dt} = v_\theta(x_t, t, s) \tag{15}$$

with the initial condition being a sample from the prior, $x_0 \sim p_0(x)$. To generate a sample, we solve this initial value problem. The solution at $t = 1$, denoted as $x_1$, is a new sample from the learned data distribution.

We also implements Classifier-Free Guidance (CFG) Ho & Salimans (2022), a technique to enhance the influence of the conditioning signal y. The effective vector field during inference becomes a weighted combination of a conditional and an unconditional prediction:

$$\hat{v}_\theta(x_t, t, s) = (1 - w) \cdot v_\theta(x_t, t) + w \cdot v_\theta(x_t, t, s) \tag{16}$$

where $w$ is the guidance weights. This allows for controlling the trade-off between sample diversity and fidelity to the conditioning information.

## 4 RESULTS

In this section, we comprehensively evaluate SAVE across multiple tasks. In Section 4.1, we begin with the introduction of the experiment setting and the implementation details. In Section 4.2, we compare it with baseline models on both single-condition and multi-condition generation across three datasets, using distributional visualization for qualitative assessment and distance-based metrics to quantify similarity between real and generated cells. We then assess SAVE's performance on two classical benchmarks: batch effect removal (Section 4.3) and out-of-sample perturbation prediction (Section 4.4). Finally, we conduct ablation and robustness studies (Section 4.5) to evaluate the contribution of each model component.

### 4.1 EXPERIMENT SETUP

Following standard bioinformatics protocols Wolf et al. (2018), the expression values for each cell were normalized to a total of $10^4$ counts and then log-transformed. Before being input into the neural network, the scRNA-seq data underwent max-absolute normalization, scaling all values to the range $[0, 1]$. The SAVE model was configured with a gene block size of $K = 3200$. The condition masking ratio was set to 0.6. Model optimization was performed using the AdamW optimizer with a learning rate of $1 \times 10^{-4}$ and a weight decay of $2.5 \times 10^{-5}$. All experiments were conducted on a GeForce RTX 3090 24GB GPU. The same set of hyperparameters was used for all experimental settings. A detailed list of model parameters is provided in Appendix Table 9.

## 4.2 CONDITIONAL GENERATION PERFORMANCE

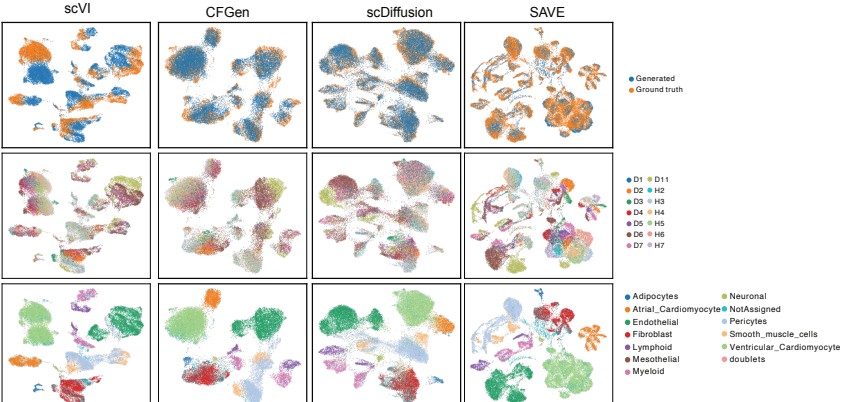

Figure 3: UMAP visualization of generative model outputs on the Heart dataset. The top row illustrates the discrepancy between the generated and real data distributions, the middle row highlights different batches, and the bottom row denotes cell types.

Table 1: Single conditional generation performance. Best results are **bolded**, and second-best are underlined.

| Method | PBMC3K | | Dentate gyrus | | Tabula Muris | |
|---|---|---|---|---|---|---|
| | WD ($\downarrow$) | MMD ($\downarrow$) | WD ($\downarrow$) | MMD ($\downarrow$) | WD ($\downarrow$) | MMD ($\downarrow$) |
| scVI | 17.66 | 0.94 | 22.61 | 1.15 | 9.76 | 0.26 |
| scDiffusion | 22.41 | 1.27 | 22.56 | 1.22 | 7.89 | 0.24 |
| CFGen | **16.94** | **0.85** | 21.55 | 1.12 | **7.39** | 0.19 |
| SAVE | 19.14 | 1.80 | **9.16** | **0.17** | 10.86 | **0.04** |

Conditional generation scenarios are categorized into three types: single-condition, dual-condition multi-platform sequencing data, and large-scale datasets with diverse expert annotations (e.g., cellx-gene Megill et al. (2021)). We compared our model with scVI, CFGen, and scDiffusion.

**Performance on single condition datasets.** We first conducted comparative experiments on datasets containing a single condition: PBMC3K[1] (conditioned on cell type), Dentate gyrus La Manno et al. (2018) (conditioned on clusters), and Tabula Muris Consortium (2020) (conditioned on tissue). As shown in Table 1, while baseline models such as CFGen perform competitively on the relatively simple PBMC3K dataset, SAVE demonstrates a significant advantage as the data complexity increases. Notably, on the Dentate gyrus dataset, SAVE dramatically outperforms all baselines, reducing both WD and MMD by more than half compared to the second-best method. On the Tabula Muris dataset, SAVE achieves the lowest MMD (0.04), indicating a highly accurate alignment with the global means of the real distribution. Although CFGen and scDiffusion exhibit strong WD scores on certain metrics, their performance fluctuates across different data scales. In contrast, SAVE provides a much more robust distributional fit for complex cellular data, demonstrating that adopting gene block attention effectively captures the overall structural dependencies that simpler architectures struggle to model.

**Performance on dual-condition datasets.** We evaluated the generative models on complex multi-platform datasets conditioned on both batch and cell type, requiring the models to simultaneously disentangle and learn the influence of two covariates. Table 2 presents the quantitative results for three representative datasets (Heart Litviňuková et al. (2020), PBMC Fischer et al. (2021), and Lung Atlas Luecken et al. (2022)). As shown in Table 2, SAVE consistently exhibits leading performance across all three evaluated datasets, achieving the lowest WD and MMD scores and demonstrating strong scalability. scDiffusion shows competitive yet second-best results. CFGen, despite employing advanced flow matching, struggles with the dual-condition complexity, performing similarly to the baseline scVI on the Lung Atlas dataset. UMAP visualizations on the Heart dataset are presented in Figure 3. While CFGen and scDiffusion demonstrate good fitting for the overall data distribution,

---

[1]https://satijalab.org/seurat/articles/pbmc3k_tutorial.html

they exhibit relatively disorganized distributions for the multi-batch Ventricular Cardiomyocyte population. In contrast, our method, SAVE, effectively distinguishes batch differences within this cell type, enabling much more accurate topological modeling.

Table 2: Dual condition generation performance.

| Method | Heart | | PBMC | | Lung Atlas | |
|---|---|---|---|---|---|---|
| | WD ($\downarrow$) | MMD ($\downarrow$) | WD ($\downarrow$) | MMD ($\downarrow$) | WD ($\downarrow$) | MMD ($\downarrow$) |
| scVI | 19.18 | 1.19 | 17.75 | 0.95 | 22.20 | 2.27 |
| CFGen | 12.57 | 0.66 | 17.41 | 0.65 | 28.02 | 1.76 |
| scDiffusion | 20.82 | 0.94 | 11.38 | 0.48 | 13.89 | 1.71 |
| SAVE | **8.30** | **0.63** | **5.37** | **0.29** | **4.37** | **1.14** |

**Multi-condition generation performance.** To evaluate the fitting of complex multi-conditional distributions, we utilized a Lung Cancer dataset Salcher et al. (2022) characterized by five conditions: sequencing protocol, developmental stage, cancer status, cancer stage, and cell type. These conditions were mapped to 27 discrete types (see Appendix Table 10). Conditions 13 and 24, representing relatively smaller data subsets, were selected as the unseen test set to evaluate conditional extrapolation, while the remaining conditions formed the training set. As shown in Table 15, SAVE demonstrates robust performance in capturing the complex data distribution. For the seen conditions, SAVE achieves the best WD score and matches scDiffusion on MMD. Crucially, on the held-out unseen data, SAVE maintains the lowest WD score, indicating that its generated global distribution remains closest to the ground truth even under novel condition combinations. While scDiffusion achieves a slightly lower MMD on the unseen set, SAVE provides a more balanced and accurate overall fit, as further supported by the visual alignments in Figure 4. Figure 4(a) demonstrates that even with limited training data, CFGen and scDiffusion generate data distributions that deviate significantly from the real data, whereas SAVE produces a much more precise distribution. Figure 4(b) further highlights that SAVE's generated data maintains a tighter alignment with the ground truth in the latent space.

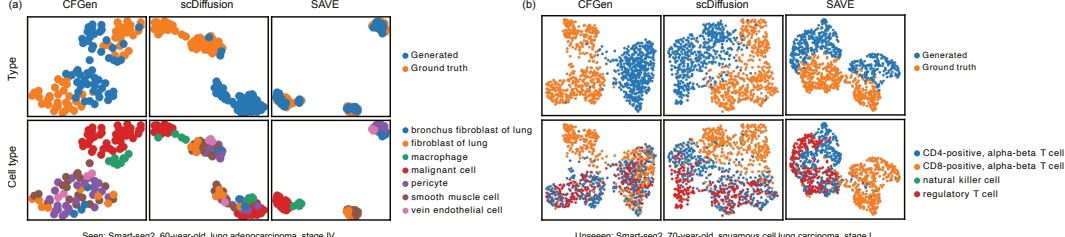

Figure 4: UMAP visualization of generative model performance on the Lung Cancer dataset. Panel (a) illustrates results on the seen conditions, and panel (b) shows performance on the unseen conditions.

Table 3: Quantitative evaluation of generative model performance on the Lung Cancer dataset. WD and MMD for seen and unseen conditions are averaged.

| Method | Seen | | Unseen | |
|---|---|---|---|---|
| | WD ($\downarrow$) | MMD ($\downarrow$) | WD ($\downarrow$) | MMD ($\downarrow$) |
| CFGen | $16.94 \pm 4.19$ | $1.49 \pm 1.80$ | $23.67 \pm 3.08$ | $2.76 \pm 2.64$ |
| scDiffusion | $5.27 \pm 2.14$ | $1.06 \pm 1.59$ | $5.29 \pm 2.03$ | **$1.87 \pm 2.31$** |
| SAVE | **$3.10 \pm 0.96$** | **$1.06 \pm 1.48$** | **$4.63 \pm 0.95$** | $2.11 \pm 2.11$ |

## 4.3 BATCH EFFECT CORRECTION

Batch effect correction is commonly employed to align multiple sequencing datasets, mitigating its impact on downstream analyses such as differential gene expression and gene regulatory network inference. Its effectiveness is typically assessed using metrics such as the biological conservation score (Bio.) and the batch correction score (Batch). The scIB score provides a comprehensive evaluation by jointly considering both biological preservation and batch mixing performance. We

systematically evaluated SAVE on three multi-platform scRNA-seq datasets (used in dual-condition generation task) and compared its batch correction performance against four established methods: Scanorama Hie et al. (2019), Harmony Korsunsky et al. (2019), scVI, and trVAE. Across all datasets, SAVE consistently achieved the best overall integration quality, attaining the top biological conservation score in three datasets and the best batch correction score in two, shown in Table 4. Traditional methods like Harmony remained competitive, ranking top in batch correction for Heart and showing stable performance overall. In contrast, deep learning methods exhibited trade-offs: scVI often underperformed in biological conservation, while trVAE achieved strong batch correction but at the cost of biological fidelity. These results highlight SAVE's ability to effectively disentangle biological variation from technical noise, particularly in complex or heterogeneous tissue datasets.

Table 4: Batch effect correction performance of SAVE model.

| Method | Lung Atlas | | | Heart | | | PBMC | | |
|---|---|---|---|---|---|---|---|---|---|
| | Bio. (↑) | Batch (↑) | scIB (↑) | Bio. (↑) | Batch (↑) | scIB (↑) | Bio. (↑) | Batch (↑) | scIB (↑) |
| Scanorama | 0.70 | 0.88 | 0.77 | 0.72 | 0.82 | 0.76 | 0.71 | 0.93 | 0.80 |
| Harmony | 0.65 | **0.93** | 0.76 | **0.76** | **0.89** | **0.81** | 0.71 | **0.95** | 0.80 |
| scVI | 0.58 | 0.83 | 0.68 | 0.66 | 0.81 | 0.72 | 0.47 | 0.78 | 0.59 |
| trVAE | 0.69 | 0.90 | 0.78 | 0.75 | 0.83 | 0.78 | **0.75** | 0.91 | 0.82 |
| SAVE | **0.73** | **0.93** | **0.81** | **0.76** | 0.86 | 0.80 | **0.75** | **0.95** | **0.83** |

## 4.4 PERTURBATION PREDICTION PERFORMANCE

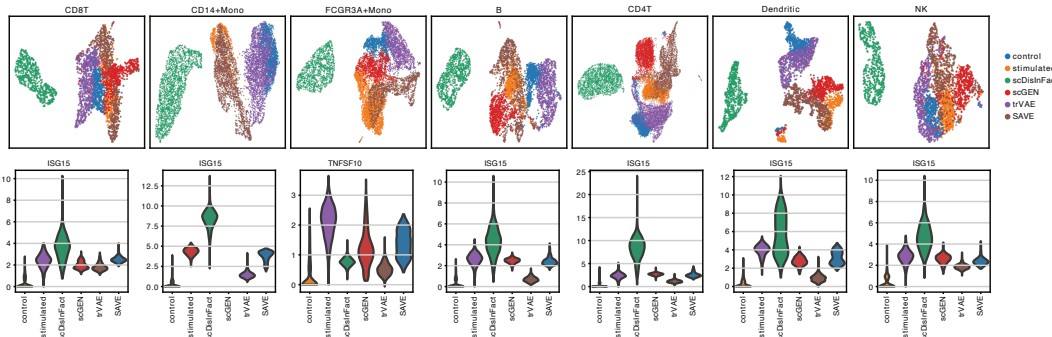

Figure 5: Performance comparison between predicted and stimulated (real perturbed) data. The top row presents UMAP visualizations comparing each method's predictions to the real perturbed condition (yellow), where closer proximity indicates better prediction accuracy. The bottom row shows violin plots of the expression distributions for the most significant differentially expressed genes predicted by each method.

Drug perturbation prediction typically involves learning the effects of drug perturbations on various cell types from training data and generalizing to novel cell types. We evaluate our approach on predicting IFN-$\beta$ drug perturbation data from the PBMC-IFN dataset Haber et al. (2017). As shown in Figure 5, SAVE produces predictions that most closely match the true stimulated (real perturbed) conditions, substantially outperforming other models. Among the baselines, scGEN demonstrates highly competitive performance, with their generated distributions tracking closely to the stimulated targets. In contrast, trVAE's outputs tend to resemble the control condition, suggesting limited capacity to model perturbation-specific effects. Meanwhile, scDisInFact exhibits significant deviations, likely due to optimization challenges or an emphasis on latent space alignment at the expense of accurately capturing the observed data distribution. The violin plot depicts the overall distribution of highly variable genes, and similar conclusions can be drawn.

From the perspective of quantitative analysis in Table 5, SAVE achieves an average PCC exceeding 0.95 and an $R^2$ of 0.86, consistently outperforming all baselines. CellOT Bunne et al. (2023) and scGEN serve as the strongest baselines, both achieving an average PCC of 0.94, while trVAE shows moderate performance. Conversely, MFM and scDisInFact perform the worst; notably, MFM Atanackovic et al. (2025) yields predictions (average PCC 0.71) that are inferior to even the

baseline correlation between the unperturbed control and perturbed data (average PCC 0.86). We observe that for cell types with significant differences between control and perturbed conditions, such as CD14+Mono and FCGR3A+Mono, SAVE yields superior prediction performance. Similar conclusions are drawn from the $R^2$ metric, indicating that SAVE is able to capture more conditional effects on expression values.

Table 5: Quantitative evaluation of perturbation prediction methods on the PBMC-IFN dataset.

| Method | CD8T | | CD14+Mono | | FCGR3A+Mono | | B | | CD4T | | Dendritic | | NK | | Average | |
|---|---|---|---|---|---|---|---|---|---|---|---|---|---|---|---|---|
| | PCC (↑) | $R^2$ (↑) | PCC (↑) | $R^2$ (↑) | PCC (↑) | $R^2$ (↑) | PCC (↑) | $R^2$ (↑) | PCC (↑) | $R^2$ (↑) | PCC (↑) | $R^2$ (↑) | PCC (↑) | $R^2$ (↑) | PCC (↑) | $R^2$ (↑) |
| control | 0.94 | 0.88 | 0.75 | 0.32 | 0.78 | 0.38 | 0.91 | 0.81 | 0.94 | 0.87 | 0.79 | 0.51 | 0.93 | 0.84 | 0.86 | 0.66 |
| trVAE | 0.98 | 0.96 | 0.82 | 0.39 | 0.83 | 0.32 | 0.90 | 0.78 | 0.97 | 0.94 | 0.83 | 0.46 | 0.97 | 0.92 | 0.90 | 0.68 |
| scDisInFact | 0.85 | 0.43 | 0.77 | 0.47 | 0.71 | 0.43 | 0.84 | 0.45 | 0.85 | 0.43 | 0.76 | 0.47 | 0.79 | 0.44 | 0.80 | 0.45 |
| scGEN | 0.96 | 0.88 | 0.95 | 0.80 | 0.93 | 0.77 | 0.91 | 0.83 | 0.92 | 0.85 | 0.96 | 0.89 | 0.93 | 0.84 | 0.94 | 0.84 |
| MFM | 0.78 | 0.60 | 0.77 | 0.53 | 0.56 | 0.08 | 0.70 | 0.49 | 0.70 | 0.49 | 0.72 | 0.51 | 0.72 | 0.51 | 0.71 | 0.46 |
| CellOT | 0.99 | 0.97 | 0.99 | 0.98 | 0.77 | 0.02 | 0.95 | 0.89 | 0.97 | 0.94 | 0.96 | 0.88 | 0.96 | 0.90 | 0.94 | 0.80 |
| SAVE | 0.98 | 0.97 | 0.97 | 0.89 | 0.91 | 0.53 | 0.97 | 0.94 | 0.96 | 0.97 | 0.96 | 0.81 | 0.96 | 0.90 | 0.96 | 0.86 |

## 4.5 ABLATION

**Ablation of gene block attention.** As shown in Table 6, removing this module leads to a drop in performance, particularly on the WD metric. This suggests that gene block attention helps the model capture structured relationships among genes and learn biologically meaningful representations.

**Effect of gene block size.** Table 7 shows the impact of block size $K$ on generation, where a suitable $K$ optimizes both MMD and WD. Separately, Table 8 evaluates training efficiency under a fixed 24GB VRAM limit. Compared to naive attention ($K = 1$), applying block attention delivers a $191\times$ speedup (e.g., 12.5 vs. 2391.2 mins), drastically reducing computational costs.

Table 6: Ablation study of Attention module.

| Setting | WD (↓) | MMD (↓) |
|---|---|---|
| SAVE w/o Att. | 8.89 | 0.65 |
| SAVE | **8.30** | **0.63** |

Table 7: Ablation study of gene block size.

| $K$ | $L$ | WD (↓) | MMD (↓) |
|---|---|---|---|
| 600 | 32 | 9.70 | 0.66 |
| 1600 | 12 | 9.64 | 0.65 |
| 2400 | 8 | 8.64 | **0.62** |
| 3200 | 6 | **8.30** | 0.63 |
| 4000 | 5 | 8.37 | 0.63 |
| 5600 | 4 | 8.41 | 0.63 |

Table 8: Training times use different block size.

| $K$ | $L$ | Time (min) |
|---|---|---|
| 1 | 19112 | 2391.2 |
| 1600 | 12 | 16.4 |
| 3200 | 6 | 12.5 |
| 5600 | 4 | 9.0 |

## 5 CONCLUSION AND LIMITATION

We present SAVE, a unified generative framework for multi-condition single-cell modeling. By integrating a generative model with a conditional Transformer and incorporating knowledge-inspired gene block attention with masked condition modeling, SAVE can generate realistic cell profiles across diverse—even unseen—conditions. Experiments on public datasets show that SAVE consistently outperforms existing methods in conditional generation, batch correction, and perturbation prediction, with strong generalization in low-data and novel-condition settings. Overall, SAVE provides a versatile tool for virtual cell synthesis and single-cell analysis, advancing generalizable and biologically grounded generative modeling.

Although our LLM-based gene block construction effectively captures high-order semantic relationships, it inherently relies on the comprehensiveness of existing literature and database annotations. While this data-driven approach yields rich functional context for well-studied genes, it may produce less informative or noisy embeddings for poorly characterized or newly discovered genes due to sparse text descriptions. Furthermore, in contrast to manually curated, gold-standard pathway databases (e.g., MSigDB or KEGG) that provide rigorously validated biological interactions, our unsupervised text-driven grouping is more susceptible to historical literature bias. Future iterations of the SAVE framework could mitigate this by integrating structured biological knowledge graphs or curated gene regulatory networks with the LLM embeddings.

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

## A    CODE OF SAVE MODEL

The main code is available at https://github.com/fdu-wangfeilab/sc-save.

## B    FUNCTIONAL VALIDATION THROUGH CONDITION-MODULATED BLOCK ATTENTION

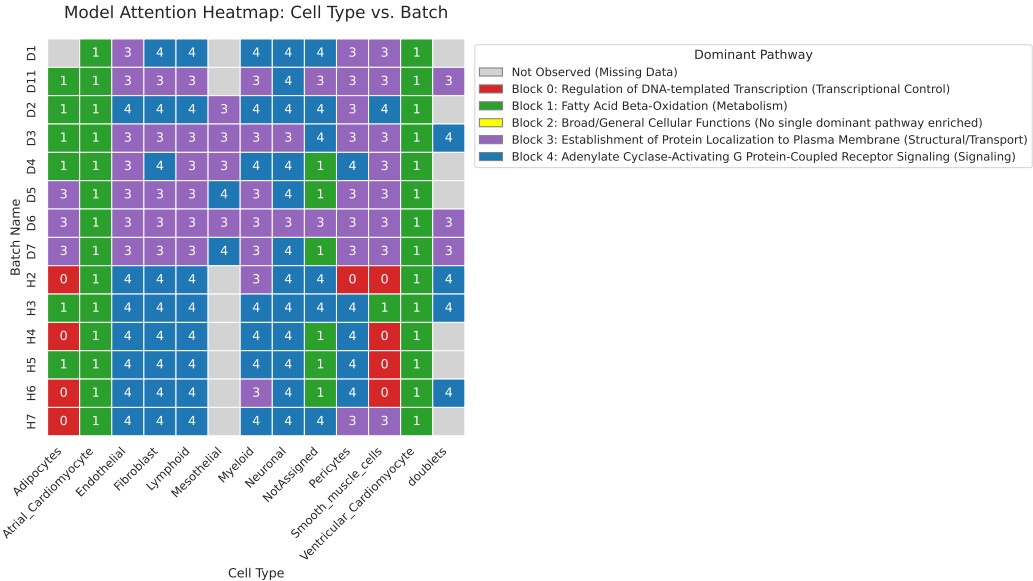

Figure 6: Condition-Modulated Attention Heatmap across Cell Types and Batches.

To ensure the creation of balanced semantic groups, we initialized our approach by selecting 16,000 highly variable genes (HVGs) from the Heart dataset. These were partitioned into five distinct blocks, each having a fixed size of 3,200 genes. To rigorously define the biological identity of each block, we identified the top-50 "centroid genes" (those closest to the embedding cluster center) and performed Gene Ontology (GO) enrichment analysis. As shown in Table 1, this partitioning successfully resulted in blocks corresponding to distinct and specific biological pathways, ranging from Metabolism (Block 1) to GPCR Signaling (Block 4).

To further validate that the model dynamically leverages these defined biological meanings, we analyzed the condition-modulated attention mechanism within the Flow Matching transformer. Specifically, we extracted and averaged the attention weights from the final layer across all four heads to pinpoint the gene block that was "most attended" under specific biological conditions.

The resulting attention heatmap (Figure 6) reveals that our model transcends simple statistical fitting and successfully captures canonical biological semantics:

- Physiological Alignment: The model accurately identifies the distinct metabolic signature of Cardiomyocytes (both Atrial and Ventricular) by exclusively attending to Block 1 (Fatty Acid Beta-Oxidation). This mirrors the heart's well-established reliance on lipids as its primary energy source.

- Functional Specificity: In cell types requiring extensive environmental interaction and signaling—such as Endothelial cells and Fibroblasts—the attention dynamically shifts to Block 4 (GPCR Signaling) and Block 3 (Protein Localization), aligning precisely with their known roles in transducing extracellular signals.

- Biological Filtering: Most critically, Block 2 (Broad/General Cellular Functions) is universally suppressed across the entire heatmap. This demonstrates that the attention mechanism acts as an effective biological filter, autonomously learning to ignore uninformative gene groups while prioritizing functional modules for accurate generation.

## C  HYPERPARAMETER OF SAVE MODEL

Table 9: Model Hyperparameter Configuration

|          |                | |
|----------|----------------|--------|
| Training | optimizer      | AdamW  |
|          | lr             | 1e-4   |
|          | batch size     | 1024   |
|          | weight decay   | 2.5e-5 |
|          | epoch          | 200    |
|          | warmup epoch   | 50     |
| Model    | gene block size | 3200  |
|          | $e$            | 512    |
|          | $d$            | 64     |
|          | num_dec_block  | 3      |
|          | num_enc_block  | 3      |
|          | attention head | 4      |

## D  DETAILED CONDITION DEFINITION FOR THE LUNG CANCER DATASET

Table 10: Detailed conditions of the Cancer dataset in the multi-condition experiment. "count" represents the number of cells for each corresponding condition. The Cyan background indicates the test conditions.

| index | assay | development_stage | disease | uicc_stage | count |
|-------|-------|-------------------|---------|------------|-------|
| 0 | 10x 3' v2 | 55-year-old human stage | chronic obstructive pulmonary disease | non-cancer | 5177 |
| 1 | GEXSCOPE technology | 55-year-old human stage | lung adenocarcinoma | III or IV | 965 |
| 2 | Smart-seq2 | 55-year-old human stage | lung adenocarcinoma | IV | 2240 |
| 3 | 10x 3' v2 | 55-year-old human stage | non-small cell lung carcinoma | II | 11151 |
| 4 | 10x 3' v2 | 55-year-old human stage | normal | non-cancer | 3143 |
| 5 | BD Rhapsody Whole Transcriptome Analysis | 55-year-old human stage | squamous cell lung carcinoma | III | 8179 |
| 6 | GEXSCOPE technology | 55-year-old human stage | squamous cell lung carcinoma | III or IV | 180 |
| 7 | 10x 3' v2 | 60-year-old human stage | chronic obstructive pulmonary disease | non-cancer | 2418 |
| 8 | Smart-seq2 | 60-year-old human stage | lung adenocarcinoma | IV | 54 |
| 9 | 10x 3' v2 | 60-year-old human stage | non-small cell lung carcinoma | I | 10660 |
| 10 | 10x 3' v2 | 60-year-old human stage | squamous cell lung carcinoma | I | 4497 |
| 11 | BD Rhapsody Whole Transcriptome Analysis | 60-year-old human stage | squamous cell lung carcinoma | I | 3663 |
| 12 | GEXSCOPE technology | 60-year-old human stage | squamous cell lung carcinoma | III or IV | 4722 |
| 13 | 10x 3' v2 | 65-year-old human stage | chronic obstructive pulmonary disease | non-cancer | 355 |
| 14 | 10x 3' v2 | 65-year-old human stage | lung adenocarcinoma | I | 10130 |
| 15 | BD Rhapsody Whole Transcriptome Analysis | 65-year-old human stage | lung adenocarcinoma | I | 2090 |
| 16 | 10x 3' v2 | 65-year-old human stage | lung adenocarcinoma | III | 6102 |
| 17 | GEXSCOPE technology | 65-year-old human stage | lung adenocarcinoma | III or IV | 1394 |
| 18 | 10x 3' v2 | 65-year-old human stage | normal | non-cancer | 7825 |
| 19 | GEXSCOPE technology | 65-year-old human stage | squamous cell lung carcinoma | III or IV | 2592 |
| 20 | 10x 3' v2 | 70-year-old human stage | chronic obstructive pulmonary disease | non-cancer | 2064 |
| 21 | 10x 3' v2 | 70-year-old human stage | lung adenocarcinoma | I | 9 |
| 22 | BD Rhapsody Whole Transcriptome Analysis | 70-year-old human stage | lung adenocarcinoma | II | 13604 |
| 23 | BD Rhapsody Whole Transcriptome Analysis | 70-year-old human stage | squamous cell lung carcinoma | I | 9056 |
| 24 | Smart-seq2 | 70-year-old human stage | squamous cell lung carcinoma | I | 507 |
| 25 | 10x 3' v1 | 70-year-old human stage | squamous cell lung carcinoma | II | 956 |
| 26 | Smart-seq2 | 70-year-old human stage | squamous cell lung carcinoma | III | 867 |

## E  PERTURBATION PREDICTION PROCEDURE AND EVALUATION

We obtained data for evaluating perturbation prediction using a Python script available at `https://github.com/theislab/scgen-reproducibility/blob/master/code/DataDownloader.py`, with parameters 'pbmc'. The training and validation datasets for PBMC IFN-$\beta$ were merged for our evaluation.

For perturbation prediction, we employed a leave-one-out approach where each cell type was iteratively selected as the test set, with the remaining cell types serving as training data. SAVE was trained on this data, with the perturbation condition input into the AdaCond module. Post-training, SAVE predicted the stimulated test data for comparison with the true stimulated data. We assessed the similarity between average gene expressions using Coefficient of determination $R^2$,

Mean Squared Error (MSE), and Pearson Correlation Coefficient (PCC). $R^2$ and MSE were calculated using scikit-learn, while PCC was computed using SciPy.

We compared SAVE with trVAE, scDisInFact, and scGEN.

- scGEN: Utilizes vector arithmetic in the latent space of its autoencoder for perturbation response prediction. We followed the tutorial at `https://scgen.readthedocs.io/en/stable/tutorials/scgen_perturbation_prediction.html`.

- trVAE: Employs its *network.predict()* function with the perturbation condition to predict perturbation responses.

- scDisInFact: A deep learning method that disentangles latent factors to generate gene expression data. We implemented it following the tutorial at `https://github.com/ZhangLabGT/scDisInFact`.

## F    EVALUATION OF BATCH EFFECT CORRECTION

We conducted batch effect correction experiments using five multi-batch datasets: Mouse, PBMC, Pancreas, Heart, Lung. These datasets were obtained from the following sources:

- Mouse: `https://drive.google.com/file/d/1lJ1AdHsfdiQHDaaO6eG9F95USmMBXOkW/view?usp=sharing`

- PBMC: `https://drive.google.com/file/d/1oKpcxQSm238SMNY77TAR5bzKUMIqgGU_/view?usp=sharing`

- Pancreas: `https://doi.org/10.6084/m9.figshare.12420968.v8`

- Heart: `https://github.com/YosefLab/scVI-data/blob/master/hca_subsampled_20k.h5ad`

- Lung Atlas: `https://doi.org/10.6084/m9.figshare.12420968.v8`

To evaluate batch effect correction, we compared SAVE with four methods: Harmony, Scanorama, scVI, trVAE. We followed the default pipelines for each method to perform integration across all datasets. The implementation details for each method are as follows:

- scVI: The integrated molecular space profile was obtained using the *SCVI.get_normalized_expression()* function.

- Harmony: We utilized the Python package available at `https://github.com/slowkow/harmonypy`. Harmony employs a fast and effective algorithm to project various batch data into a common space.

- Scanorama*: We employed the same function used for latent integration, *scanorama.correct_scanpy(adatas, return_dimred=True)*. The corrected *adatas.X* was used as the integrated molecular space profile.

- trVAE: We utilized the *network.predict()* function to transfer the input scRNA-seq data to a specific batch. For scIB evaluation, we applied this function to transfer input data to each batch and calculated the average of transferred scIB scores as the final scIB score.

For visualization, we employed the UMAP algorithm using default parameters from the Python package ScanpyWolf et al. (2018). Clustering results were annotated with the cell type and batch information from the raw data. We utilized the 50-dimensional PCA features of the corrected data for clustering.

The batch effect correction performance are evaluated using 10 well-established metrics from the scIB package with default parametersLuecken et al. (2022). We used the overall score, which is the average of batch correction and bio-information conservation performance, as the final evaluation metric.

Table 11: Ablation of condition mask on VAE perturbation prediction.

| Method | CD8T | | CD14+Mono | | FCGR3A+Mono | | B | | CD4T | | Dendritic | | NK | | Average | |
|---|---|---|---|---|---|---|---|---|---|---|---|---|---|---|---|---|
| | PCC (↑) | $R^2$ (↑) | PCC (↑) | $R^2$ (↑) | PCC (↑) | $R^2$ (↑) | PCC (↑) | $R^2$ (↑) | PCC (↑) | $R^2$ (↑) | PCC (↑) | $R^2$ (↑) | PCC (↑) | $R^2$ (↑) | PCC (↑) | $R^2$ (↑) |
| SAVE w/o cond mask | 0.91 | 0.82 | 0.92 | 0.70 | 0.92 | **0.72** | 0.89 | 0.78 | 0.89 | 0.77 | 0.91 | 0.71 | 0.86 | 0.72 | 0.90 | 0.75 |
| SAVE | **0.98** | **0.97** | **0.97** | **0.89** | **0.91** | 0.53 | **0.97** | **0.94** | **0.96** | 0.97 | **0.96** | **0.81** | **0.96** | **0.90** | **0.96** | **0.86** |

# G    ABLATION OF CONDITIONAL MASK MODELING.

To test the effectiveness of conditional mask modeling, we ablated this component by directly feeding condition labels without masking. As shown in Table 11, this led to a marked decrease in generalization performance on unseen condition combinations. The results suggest that conditional masking acts as a regularizer, encouraging the model to learn more transferable condition embeddings, which in turn improves extrapolation.

# H    GENERATION PERFORMANCE

## H.1    GENE LEVEL PERFORMANCE

Table 12: Quantitative evaluation of different models across three datasets (Best: **bold**, Second best: underline).

| Model | PBMC3k | | | Dentate | | | Muris | | |
|---|---|---|---|---|---|---|---|---|---|
| | MSE | PCC | $R^2$ | MSE | PCC | $R^2$ | MSE | PCC | $R^2$ |
| scVI | **0.07 ± 0.14** | **0.93 ± 0.12** | **0.87 ± 0.22** | **0.00 ± 0.00** | **0.99 ± 0.01** | **0.99 ± 0.01** | 0.07 ± 0.03 | 0.98 ± 0.01 | 0.30 ± 0.32 |
| CFGen | 0.09 ± 0.17 | 0.91 ± 0.14 | 0.82 ± 0.26 | **0.00 ± 0.00** | **0.99 ± 0.01** | 0.97 ± 0.02 | **0.00 ± 0.00** | **1.00 ± 0.00** | **0.99 ± 0.01** |
| scDiffusion | 0.11 ± 0.12 | 0.86 ± 0.10 | 0.73 ± 0.17 | **0.00 ± 0.00** | **0.99 ± 0.01** | 0.98 ± 0.01 | 0.01 ± 0.01 | 0.99 ± 0.01 | 0.94 ± 0.09 |
| SAVE | 0.08 ± 0.18 | **0.93 ± 0.12** | 0.84 ± 0.27 | **0.00 ± 0.00** | **0.99 ± 0.01** | 0.97 ± 0.02 | **0.00 ± 0.00** | 0.99 ± 0.01 | 0.95 ± 0.05 |

Table 13: Quantitative evaluation of different models across three new datasets (Best: **bold**, Second best: underline).

| Model | Heart | | | PBMC | | | Lung | | |
|---|---|---|---|---|---|---|---|---|---|
| | MSE | PCC | $R^2$ | MSE | PCC | $R^2$ | MSE | PCC | $R^2$ |
| scVI | 0.07 ± 0.09 | 0.84 ± 0.14 | −0.65 ± 1.45 | 0.05 ± 0.05 | 0.87 ± 0.12 | 0.70 ± 0.20 | 0.08 ± 0.07 | 0.87 ± 0.17 | −0.57 ± 0.96 |
| CFGen | 0.04 ± 0.09 | 0.87 ± 0.18 | 0.75 ± 0.32 | 0.06 ± 0.06 | 0.87 ± 0.08 | 0.75 ± 0.15 | 0.14 ± 0.15 | 0.58 ± 0.20 | 0.17 ± 0.52 |
| scDiffuion | 0.02 ± 0.04 | 0.87 ± 0.14 | **0.79 ± 0.22** | 0.03 ± 0.03 | 0.93 ± 0.06 | 0.86 ± 0.10 | 0.03 ± 0.02 | 0.65 ± 0.14 | 0.35 ± 0.24 |
| SAVE | **0.01 ± 0.02** | **0.88 ± 0.13** | 0.63 ± 0.26 | **0.02 ± 0.02** | **0.98 ± 0.03** | **0.95 ± 0.07** | **0.01 ± 0.01** | **0.91 ± 0.11** | **0.84 ± 0.18** |

## H.2    VISUALIZATION OF SINGLE CONDITION PERFORMANCE

Figures 7, 8, and 9 show the UMAP visualizations of the generative model on the single-condition datasets PBMC3K `https://satijalab.org/seurat/articles/pbmc3k_tutorial.html`, Dentate Gyrus La Manno et al. (2018), and Tabula Muris Consortium (2018), respectively.

## H.3    VISUALIZATION OF DUAL CONDITION PERFORMANCE

Figures 10, 11, 12 and 13 show the UMAP visualizations of the generative model on the dual-condition datasets Mouse endocrinogenesis, PBMC, Pancreas and Lung Atlas, respectively.

## H.4    MULTI-CONDITION GENERATION PERFORMANCE

**Setup.** For the Lung Cancer dataset Salcher et al. (2022), we selected samples corresponding to developmental ages from 55 to 75 years, at 5-year intervals. These samples were divided into 27 categories, each containing heterogeneous cell types, yielding a total of 618 unique conditions. We designated conditions 13 and 24, characterized by smaller data volumes, as the test set, while the remaining data served as the training set. Five condition categories were utilized: assay, development_stage, disease, uicc_stage, and cell type. Given that scVI is not designed to handle unknown conditions, our comparison here is restricted to CFGen, scDiffusion, and SAVE. CFGen uses classifier-free guidance for condition combination, whereas scDiffusion employs classifier guidance.

Table 14: Quantitative evaluation of models on Seen and Unseen datasets (Best: **bold**, Second best: underline).

| Model | Seen | | | Unseen | | |
|---|---|---|---|---|---|---|
| | MSE | PCC | $R^2$ | MSE | PCC | $R^2$ |
| CFGen | $0.28 \pm 0.25$ | $0.73 \pm 0.17$ | $0.48 \pm 0.33$ | $0.62 \pm 0.41$ | $0.56 \pm 0.16$ | $0.20 \pm 0.30$ |
| scDiffusion | $\underline{0.03 \pm 0.03}$ | $\underline{0.85 \pm 0.14}$ | $\underline{0.72 \pm 0.27}$ | $\underline{0.05 \pm 0.05}$ | $\underline{0.81 \pm 0.14}$ | $\underline{0.63 \pm 0.27}$ |
| SAVE | $\mathbf{0.01 \pm 0.02}$ | $\mathbf{0.94 \pm 0.07}$ | $\mathbf{0.88 \pm 0.14}$ | $\mathbf{0.04 \pm 0.03}$ | $\mathbf{0.85 \pm 0.08}$ | $\mathbf{0.70 \pm 0.14}$ |

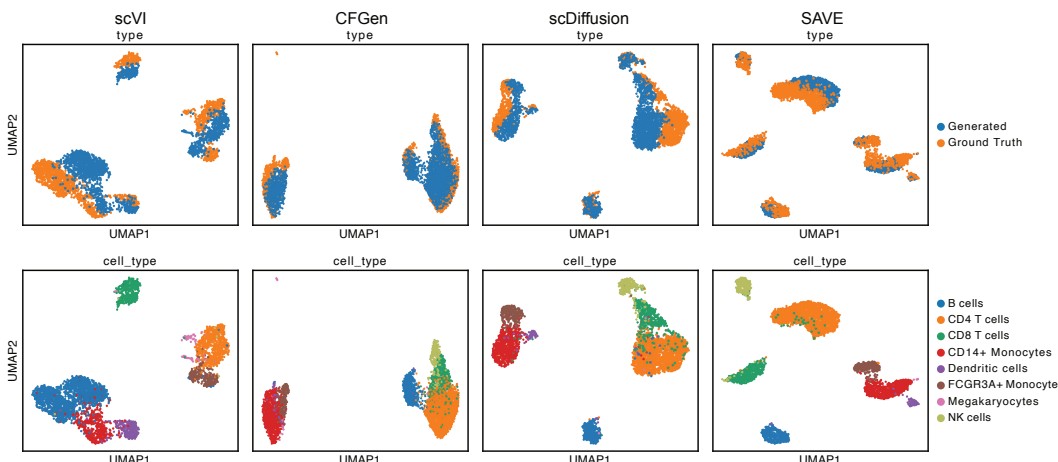

Figure 7: Generation results of the generative models on the PBMC3K dataset, visualized using UMAP.

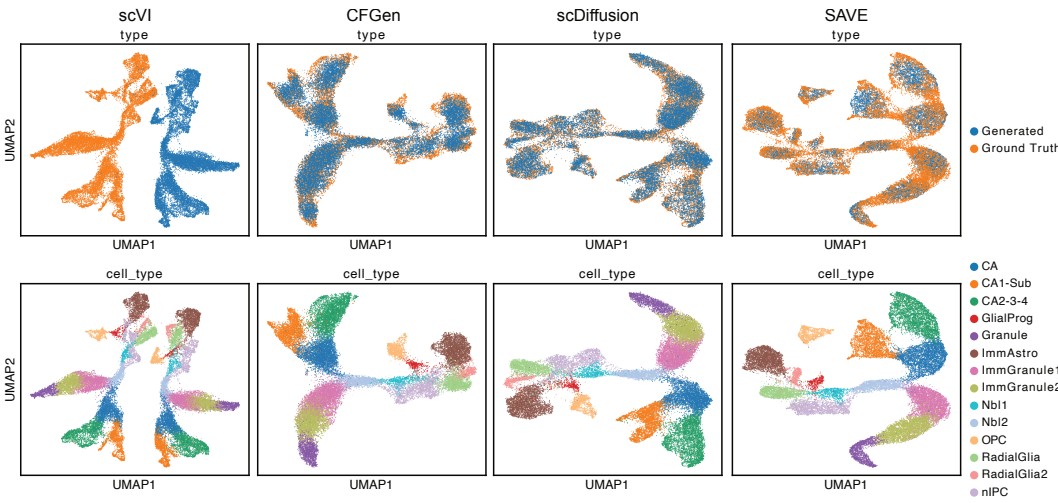

Figure 8: Generation results of the generative models on the Dentate gyrus dataset, visualized using UMAP.

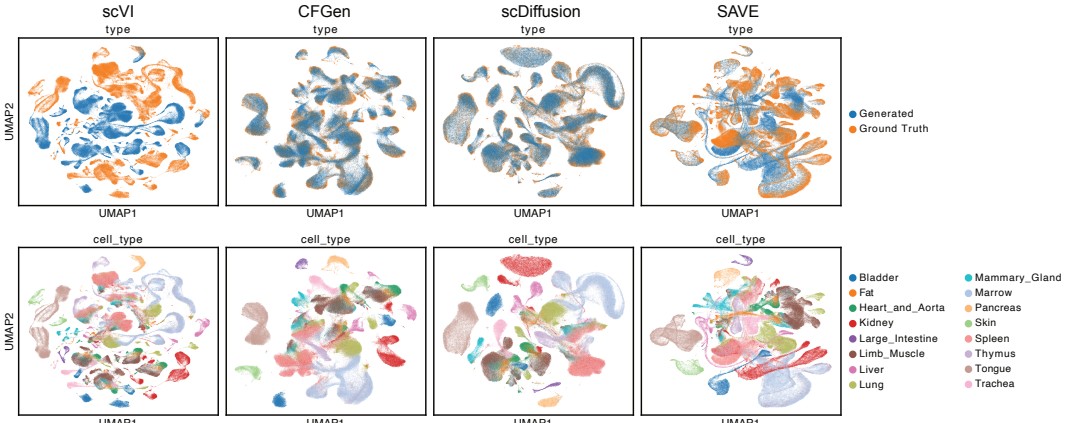

Figure 9: Generation results of the generative models on the Tabula Muris dataset, visualized using UMAP.

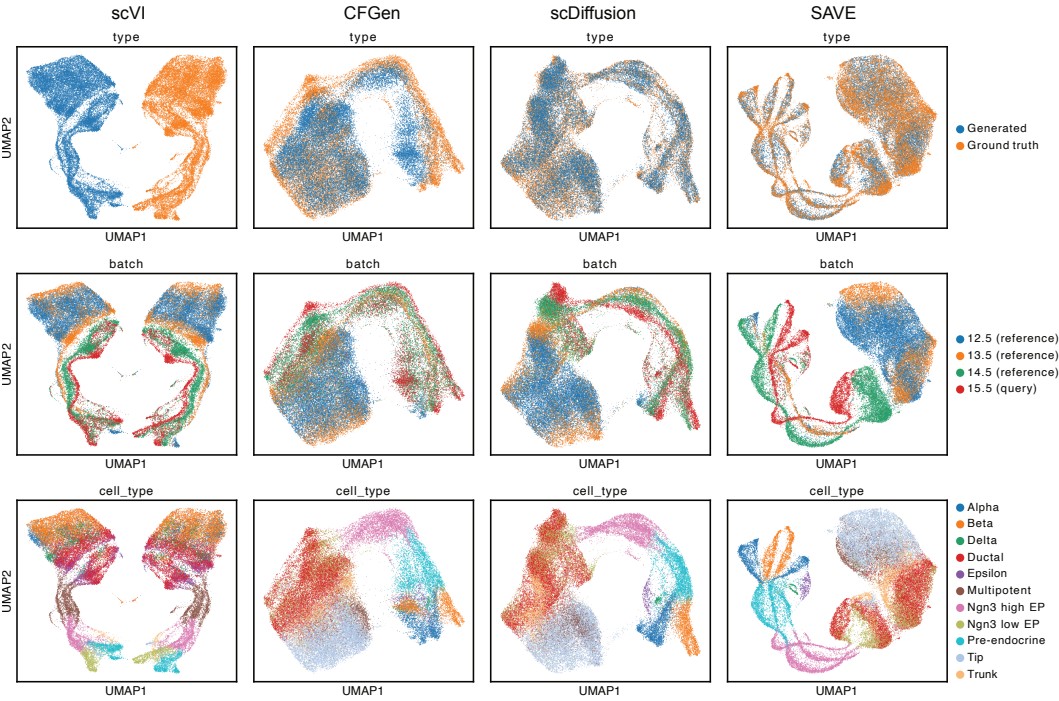

Figure 10: Generation results of the generative models on the Mouse endocrinogenesis dataset, visualized using UMAP.

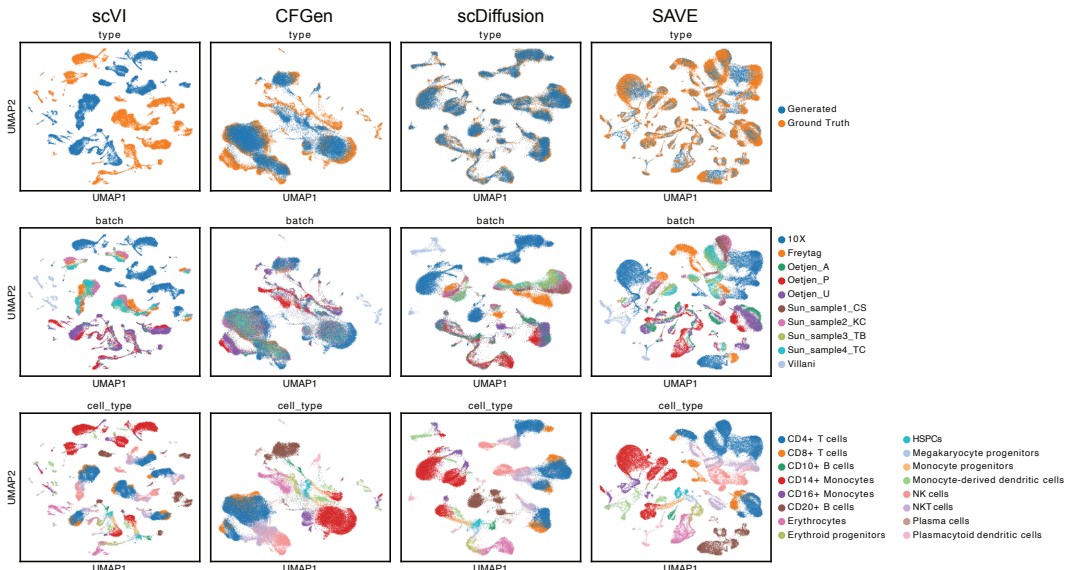

Figure 11: Generation results of the generative models on the PBMC dataset, visualized using UMAP.

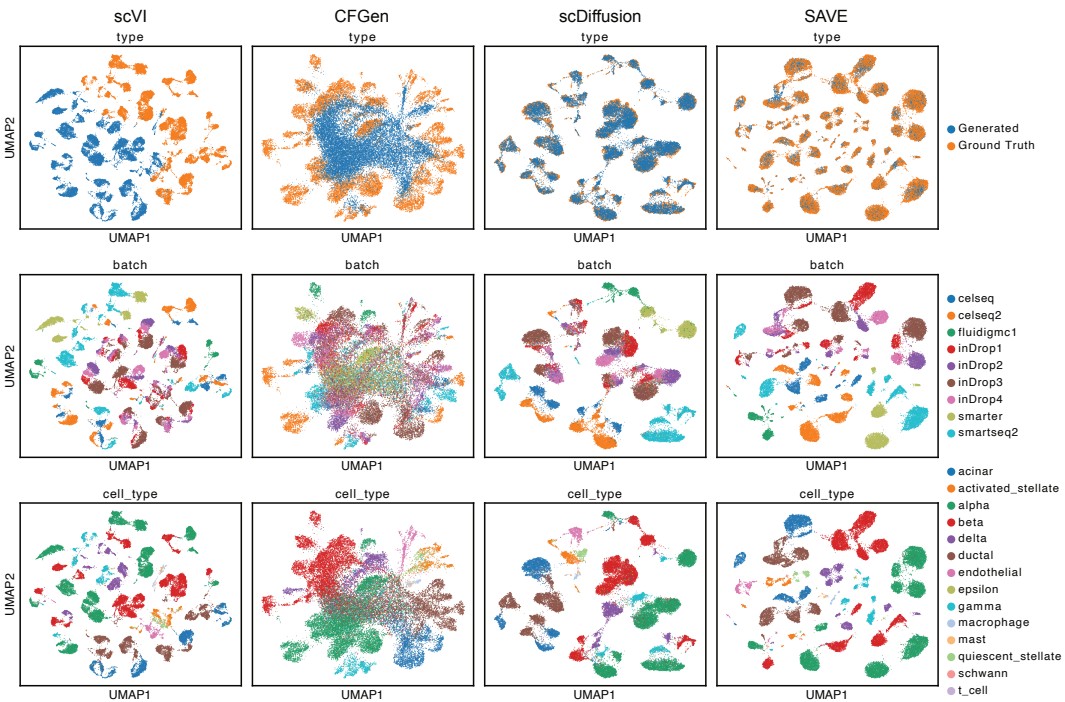

Figure 12: Generation results of the generative models on the Pancreas dataset, visualized using UMAP.

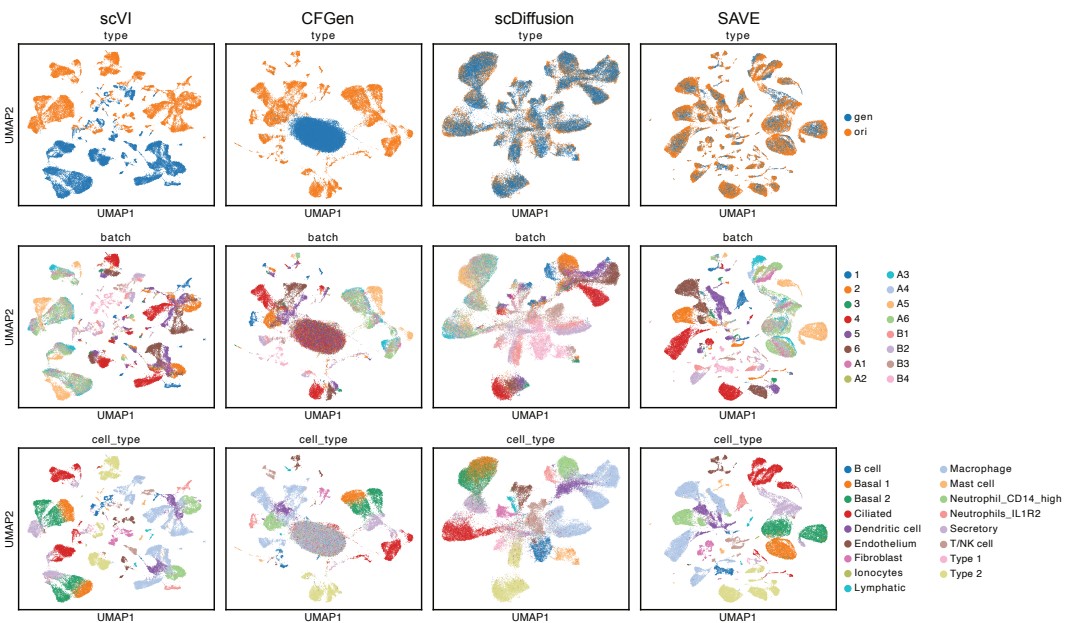

Figure 13: Generation results of the generative models on the Lung Atlas, visualized using UMAP.

**Results.** We showcase the performance of the generative model on the multi-condition lung cancer dataset in Table 15. We also provide comparisons with ablation studies that omit the attention mechanism and the latent mask. The SAVE model consistently demonstrates superior performance over other compared methods.

# I  BATCH EFFECT CORRECTION

In this section, we present detailed metrics for batch effect correction. These encompass biological information preservation metrics Adjusted Rand Index (ARI) Normalized mutual information (NMI), Average silhouette width (ASW) and batch effect removal metrics ASW batch, Principal component regression score (PCR), Graph Connectivity (graph conn). Detailed metrics can be found at `https://scib.readthedocs.io/en/latest/api.html#batch-correction-metrics`. SAVE demonstrates good performance across all these metrics.

# J  PERTURBATION PREDICTION PERFORMANCE

## J.1  WD AND MMD OF PERTURBATION PERFORMANCE

## J.2  MSE AND DISTRIBUTION OF TOP 5 DEGS ON PBMC IFN-$\beta$

In Table 20, we present the Mean Squared Error (MSE) between the gene expression of predictions from perturbation prediction methods and those of the ground truth on PBMC IFN-$\beta$. In Figure 14, we show the distribution of the top 5 differentially expressed genes (DEGs) predicted by perturbation prediction methods. On both evaluation metrics, SAVE's prediction is closest to the ground truth.

Table 15: Performance of generative models for each condition on the Lung Cancer dataset.

| Condition | CFGen | | scDiffusion | | SAVE | |
|---|---|---|---|---|---|---|
| | WD ($\downarrow$) | MMD ($\downarrow$) | WD ($\downarrow$) | MMD ($\downarrow$) | WD ($\downarrow$) | MMD ($\downarrow$) |
| 0 | 20.89 | 1.34 | 4.87 | 0.63 | 2.66 | 0.67 |
| 1 | 15.36 | 1.86 | 5.33 | 1.45 | 3.24 | 1.5 |
| 2 | 16.08 | 1.04 | 10.31 | 1.2 | 2.54 | 0.75 |
| 3 | 13.29 | 0.68 | 3.76 | 0.37 | 2.56 | 0.52 |
| 4 | 17.97 | 1.5 | 8.47 | 1.24 | 6.47 | 0.97 |
| 5 | 16.09 | 1.03 | 4.97 | 0.66 | 2.97 | 0.77 |
| 6 | 15.65 | 2.51 | 6.95 | 2.12 | 4.1 | 1.97 |
| 7 | 23.28 | 1.27 | 4.38 | 0.46 | 2.66 | 0.59 |
| 8 | 14.49 | 2.13 | 11.76 | 2.38 | 3.35 | 1.68 |
| 9 | 12.8 | 0.53 | 4.51 | 0.31 | 2.59 | 0.43 |
| 10 | 13.36 | 0.83 | 4.43 | 0.55 | 2.54 | 0.69 |
| 11 | 16.75 | 1.49 | 4.68 | 1.04 | 2.85 | 1.12 |
| 12 | 19.6 | 2.73 | 7.91 | 2.22 | 3.42 | 1.97 |
| 13 | 24.15 | 2.83 | 5.39 | 1.9 | 4.6 | 2.15 |
| 14 | 13.34 | 1.67 | 4.12 | 1.28 | 2.88 | 1.36 |
| 15 | 14.96 | 1.26 | 4.87 | 0.89 | 2.8 | 0.98 |
| 16 | 14.51 | 0.73 | 4.63 | 0.44 | 2.8 | 0.55 |
| 17 | 18.47 | 3.53 | 5.87 | 2.76 | 3.31 | 2.67 |
| 18 | 21.02 | 1.51 | 5 | 0.77 | 2.67 | 0.83 |
| 19 | 20.83 | 2.43 | 6.55 | 1.77 | 3.05 | 1.68 |
| 20 | 22.28 | 2.00 | 5.03 | 1.19 | 2.94 | 1.21 |
| 21 | 18.83 | 5.8 | 5.13 | 4.7 | 4.28 | 4.72 |
| 22 | 16 | 1.04 | 3.74 | 0.6 | 2.8 | 0.75 |
| 23 | 15.32 | 0.72 | 4.65 | 0.4 | 2.75 | 0.51 |
| 24 | 20.89 | 2.38 | 4.7 | 1.68 | 3.36 | 1.86 |
| 25 | 14.01 | 2.12 | 5.07 | 1.69 | 2.79 | 1.68 |
| 26 | 20.12 | 0.58 | 4.21 | 0.2 | 2.16 | 0.32 |
| Average | 17.42 | 1.76 | 5.60 | 1.29 | 3.15 | 1.29 |

Table 16: Detailed scIB scores for batch effect correction methods on the PBMC dataset.

| | NMI_cluster/label | ARI_cluster/label | ASW_label | ASW_label/batch | PCR_batch | graph_conn | avg_bio | avg_batch | scib_score |
|---|---|---|---|---|---|---|---|---|---|
| Scanorama | 0.8053 | 0.7521 | 0.5838 | 0.8989 | 0.5067 | 0.9577 | 0.7138 | 0.9283 | 0.7996 |
| Harmony | 0.8008 | 0.7605 | 0.5541 | **0.9243** | **0.6959** | 0.9686 | 0.7051 | **0.9464** | 0.8016 |
| scVI | 0.5614 | 0.2905 | 0.5594 | 0.7678 | - | 0.7855 | 0.4704 | 0.7767 | 0.5929 |
| trVAE | 0.8276 | 0.8312 | 0.6051 | 0.8487 | 0.6322 | 0.9779 | 0.7546 | 0.9133 | 0.8181 |
| SAVE | **0.8637** | **0.7952** | **0.5958** | 0.9128 | 0.4996 | **0.9868** | **0.7516** | 0.9498 | **0.8309** |

Table 17: Detailed scIB scores for batch effect correction methods on the Lung Atlas dataset.

| | NMI_cluster/label | ARI_cluster/label | ASW_label | ASW_label/batch | PCR_batch | graph_conn | avg_bio | avg_batch | scib_score |
|---|---|---|---|---|---|---|---|---|---|
| Scanorama | 0.7856 | 0.6977 | 0.6159 | 0.8241 | **0** | 0.9305 | 0.6997 | 0.8773 | 0.7708 |
| Harmony | 0.7727 | 0.5973 | 0.5741 | **0.9039** | 0.3605 | 0.9472 | 0.648 | **0.9255** | 0.759 |
| scVI | 0.7338 | 0.4599 | 0.5478 | 0.7851 | 0.0557 | 0.8846 | 0.5805 | 0.8348 | 0.6822 |
| trVAE | 0.7942 | 0.6462 | 0.6355 | 0.8265 | 0.4414 | 0.9826 | 0.6919 | 0.9046 | 0.777 |
| SAVE | **0.8285** | **0.7340** | **0.6395** | 0.8693 | 0.2134 | **0.9926** | **0.7340** | 0.9309 | **0.8128** |

Table 18: Detailed scIB scores for batch effect correction methods on the Heart dataset.

| | NMI_cluster/label | ARI_cluster/label | ASW_label | ASW_label/batch | PCR_batch | graph_conn | avg_bio | avg_batch | scib_score |
|---|---|---|---|---|---|---|---|---|---|
| Scanorama | 0.7538 | 0.7391 | 0.6653 | 0.796 | 0.2708 | 0.8438 | 0.7194 | 0.8199 | 0.7596 |
| Harmony | 0.8121 | 0.843 | 0.6126 | **0.8837** | 0.4392 | 0.9015 | 0.7559 | 0.8926 | 0.8106 |
| scVI | 0.7248 | 0.6574 | 0.5975 | 0.79 | 0.063 | 0.8361 | 0.6599 | 0.813 | 0.7212 |
| trVAE | 0.7829 | 0.804 | 0.6521 | 0.8169 | 0.1098 | 0.8487 | 0.7463 | 0.8328 | 0.7809 |
| SAVE | **0.7999** | **0.8146** | **0.6579** | 0.8576 | **0.2554** | **0.8615** | **0.7574** | **0.8596** | **0.7983** |

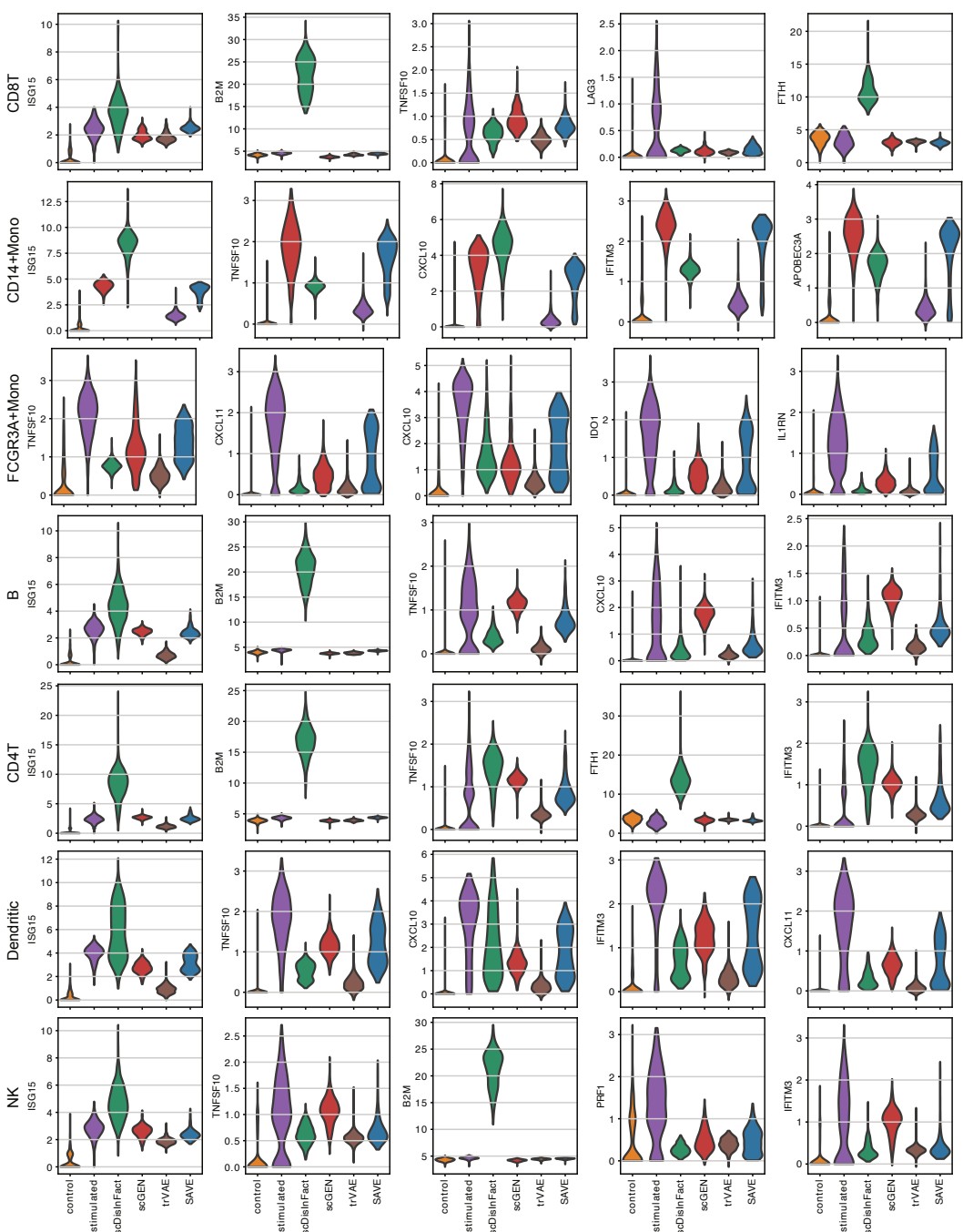

Figure 14: Violin plots of the top 5 differentially expressed genes (DEGs) predicted by perturbation prediction methods on PBMC IFN-$\beta$.

Table 19: Comparison of Wasserstein Distance (WD) and Maximum Mean Discrepancy (MMD) across different cell types and methods (Lower is better: best value in **bold**, second best underlined).

| Method | CD8T | | CD14+Mono | | FCGR3A+Mono | | B | | CD4T | | Dendritic | | NK | | Average | |
|---|---|---|---|---|---|---|---|---|---|---|---|---|---|---|---|---|
| | WD | MMD | WD | MMD | WD | MMD | WD | MMD | WD | MMD | WD | MMD | WD | MMD | WD | MMD |
| control | 4.61 | **0.07** | 9.94 | 0.54 | 9.09 | 0.44 | 4.86 | **0.10** | 3.91 | **0.07** | 8.66 | 0.36 | 5.47 | 0.10 | 6.65 | 0.24 |
| trVAE | 4.33 | 0.26 | 9.05 | 0.58 | 8.73 | 0.57 | 5.28 | 0.41 | 3.67 | 0.50 | 8.43 | 0.46 | 5.17 | 0.30 | 6.38 | 0.44 |
| scDisInFact | 20.63 | 1.40 | 23.76 | 1.49 | 22.71 | 1.37 | 18.53 | 1.31 | 19.78 | 1.36 | 21.36 | 1.18 | 19.42 | 1.35 | 20.88 | 1.35 |
| scGEN | 4.56 | 0.19 | 6.43 | 0.19 | **6.50** | **0.20** | 4.82 | 0.32 | 4.17 | 0.28 | **5.74** | **0.12** | 5.37 | 0.14 | 5.37 | 0.21 |
| MFM | 13.63 | 0.47 | 12.34 | 0.51 | 14.39 | 0.64 | 14.92 | 0.56 | 14.00 | 0.53 | 13.90 | 0.52 | 15.60 | 0.57 | 14.11 | 0.54 |
| CellOT | **3.91** | **0.07** | **4.54** | **0.05** | 9.56 | 0.54 | 4.52 | 0.18 | **3.49** | 0.12 | 6.13 | 0.13 | 4.98 | 0.13 | 5.30 | 0.17 |
| SAVE(ours) | 4.01 | 0.11 | 5.03 | 0.09 | 7.01 | 0.22 | **4.40** | 0.17 | 3.98 | 0.11 | 6.55 | 0.17 | **5.07** | **0.09** | **5.15** | **0.14** |

Table 20: Mean Squared Error (MSE) across different cell types (Lower is better: best value in **bold**, second best underlined).

| MSE | CD8T | CD14+Mono | FCGR3A+Mono | B | CD4T | Dendritic | NK | mean |
|---|---|---|---|---|---|---|---|---|
| control | **0.00** | 0.04 | 0.03 | 0.01 | **0.00** | 0.02 | 0.01 | 0.02 |
| trVAE | **0.00** | 0.03 | 0.03 | 0.01 | **0.00** | 0.02 | **0.00** | 0.01 |
| scDisInFact | 0.20 | 0.26 | 0.23 | 0.16 | 0.18 | 0.20 | 0.17 | 0.20 |
| scGEN | **0.00** | 0.01 | **0.01** | 0.01 | **0.00** | **0.01** | 0.01 | 0.01 |
| MFM | 0.02 | 0.04 | 0.07 | 0.03 | 0.03 | 0.05 | 0.04 | 0.04 |
| CellOT | **0.00** | **0.00** | 0.03 | **0.00** | **0.00** | **0.01** | **0.00** | 0.01 |
| SAVE(ours) | **0.00** | **0.00** | **0.01** | **0.00** | **0.00** | **0.01** | **0.00** | **0.00** |

