# OpenReview forum: "SAVE: A Generalizable Framework for Multi-Condition Single-Cell Generation with Gene Block Attention"
_ICLR.cc/2026/Conference — ICLR 2026 Poster_

### Official Review · Reviewer_VUna · 2025-10-29

**Soundness:** 3
**Presentation:** 3
**Contribution:** 3
**Rating:** 8
**Confidence:** 4

**Summary:**

This paper presents SAVE, a conditional generative framework for single-cell RNA sequencing (scRNA-seq) data based on a variational autoencoder with a Transformer backbone. SAVE introduces a biologically motivated gene-block attention mechanism that aggregates groups of genes into block-level tokens to capture co-regulation and pathway-level structure. Condition-specific effects are incorporated through Adaptive Layer Normalization (AdaLN), while a dual masking strategy on condition labels and latent codes enhances robustness and generalization. The framework is evaluated across three tasks—conditional single-cell generation, batch correction, and perturbation prediction and compared against established baselines including CFGen, scVI, and scDiffusion.

The paper includes a careful analysis of evaluation metrics (Wasserstein Distance, MMD, PCC, and MSE), extensive ablations on the gene-block design, and discussions of dataset- and model-specific limitations. Overall, SAVE aims to provide a unified, biologically grounded model for multi-condition single-cell data generation.

**Strengths:**

1. Biologically grounded architectural design.
The gene-block attention mechanism is an elegant way to encode biological priors into a Transformer framework. By grouping genes into functional or co-expression blocks, the model captures higher-level dependencies that would be difficult to learn from individual gene tokens. The accompanying ablation study (varying block size) provides strong empirical evidence that this inductive bias improves both performance and computational efficiency.

2. Rigorous and transparent evaluation.
The experiments are thorough and well-motivated. The authors provide detailed preprocessing pipelines, normalization strategies, and metric definitions, which help ensure reproducibility.

3. Comprehensive comparisons and analyses.
The experimental section includes meaningful comparisons to strong baselines, alongside careful discussion of potential confounding factors such as size-factor grouping and normalization space. The analyses convincingly explain discrepancies between quantitative metrics and qualitative observations.

4. Good result interpretability.
Beyond standard distributional alignment, the study evaluates gene-level fidelity and provides insight into how SAVE captures biologically relevant structures, lending credibility to its claims of biological plausibility.

**Weaknesses:**

1. Limited evaluation scope on perturbation modeling.
While the paper demonstrates SAVE’s versatility across multiple tasks, the perturbation prediction experiments are limited to a narrow set of baselines. Comparisons to more recent flow-matching or optimal-transport–based approaches would further support claims of generalization in conditional settings.

2. Decoder design justification remains empirical.
The use of a fully connected MLP decoder is empirically justified but lacks a stronger probabilistic or biological rationale. While the arguments against negative binomial modeling are sound, exploring hybrid or auxiliary likelihoods could improve interpretability without compromising stability. For example, a flow-based decoder on top of a transformer embedding is a reasonable alternative.

**Questions:**

See weaknesses.

---

> ### Author Response · Authors · 2025-11-25
>
> > W1: Limited evaluation scope on perturbation modeling. While the paper demonstrates SAVE’s versatility across multiple tasks, the perturbation prediction experiments are limited to a narrow set of baselines. Comparisons to more recent flow-matching or optimal-transport–based approaches would further support claims of generalization in conditional settings.
>
> We sincerely thank the reviewer for this constructive feedback. Please refer to our response to 16e8's **Weakness 1**, where we have addressed this by expanding our evaluation scope. We incorporated **CellOT**  (an optimal-transport-based approach) and **Meta Flow Matching (MFM)** (a recent flow-matching approach) as additional baselines. The comparative results demonstrate that **SAVE** maintains state-of-the-art performance against these advanced methods.
>
> >W2: Decoder design justification remains empirical. The use of a fully connected MLP decoder is empirically justified but lacks a stronger probabilistic or biological rationale. While the arguments against negative binomial modeling are sound, exploring hybrid or auxiliary likelihoods could improve interpretability without compromising stability. For example, a flow-based decoder on top of a transformer embedding is a reasonable alternative.
>
> We agree that this is a promising direction for future research. Currently, our decoder implementation aligns with the reconstruction methodology used in [SCALEX](https://doi.org/10.1038/s41467-022-33758-z), which ensures robust data fidelity and training stability. While we recognize the potential of exploring more diverse, biologically informed decoders (such as flow-based heads or auxiliary likelihoods), our primary focus in this work was to establish the interpretability and computational efficiency of the **Gene Block** **Attention** mechanism within the generative process. We intend to explore these alternative decoder architectures in subsequent studies.

---

### Official Review · Reviewer_a9ki · 2025-11-02

**Soundness:** 2
**Presentation:** 2
**Contribution:** 2
**Rating:** 2
**Confidence:** 3

**Summary:**

The paper proposes a unified generative framework for scRNA-seq that (i) groups genes into semantically defined gene blocks using LLM-derived gene embeddings, (ii) applies a conditional Transformer with Adaptive LayerNorm (AdaLN) to inject multi-factor conditions (batch, cell type, disease, protocol, etc.), and (iii) couples this with flow matching in latent space to enable flexible conditional generation, including unseen condition combinations.

**Strengths:**

1. The paper correctly notes that many single-cell generative models treat genes as flat tokens, ignoring biological structure and struggling with combinatorial conditions. SAVE’s gene-block idea is a plausible way to add structure.
2. Same architecture handles conditional generation, batch integration, and perturbation prediction. That’s attractive for labs that don’t want three separate models.

**Weaknesses:**

1. Gene-block construction is underspecified for reproduction. The method relies on LLM-derived gene embeddings + optimal-transport clustering to form balanced blocks, but the paper does not fully quantify: what text fields from NCBI were used, which LLM/encoder, how sensitive performance is to mistakes in gene descriptions or to species-specific vocabularies.
2. SAVE uses a VAE + Transformer over blocks + flow matching + AdaLN. That’s quite a stack. There is a note that all experiments ran on a 3090 with shared hyperparameters, but no training time, memory profile, or comparison to lighter baselines is given. For large atlases (millions of cells), this matters.
3. Evaluation axes are mostly distributional. WD/MMD and scIB show that the clouds overlap, but users often care about gene-level faithfulness for downstream DE, pathway scores, or ligand–receptor analysis. Only in the perturbation section do we see gene-level metrics (PCC, R²), extending that style of evaluation to the other datasets would strengthen the paper.
4. The paper claims “broad utility in virtual cell synthesis and biological discovery,” but doesn’t show a biological discovery enabled by SAVE.

**Questions:**

The paper adopts affine flow matching in latent space and combines it with classifier-free guidance, which is elegant, but there’s no comparison to latent diffusion or to a simple conditional ODE on top of the VAE to show why Flow Matching was preferred (stability? speed? better extrapolation?).

---

> ### Author Response · Authors · 2025-11-25
>
> > W1: Gene-block construction is underspecified for reproduction. The methzod relies on LLM-derived gene embeddings + optimal-transport clustering to form balanced blocks, but the paper does not fully quantify: what text fields from NCBI were used, which LLM/encoder, how sensitive performance is to mistakes in gene descriptions or to species-specific vocabularies.
>
>  We appreciate the reviewer pointing out the need for greater specification regarding our gene-block construction. We have clarified the implementation details and added a sensitivity analysis to the revised manuscript.
>
> **1. Implementation Details:** To construct the gene blocks, we utilized text descriptions sourced specifically from the **'Summary' section of the** **NCBI** **Gene database**. Preprocessing involved removing non-informative elements such as hyperlinks and timestamps. For vectorization, we employed OpenAI’s **`text-embedding-ada-002`** model. This encoder processed the curated gene summaries (which averaged 73 words in length) to produce 1,536-dimensional embeddings, thereby capturing biological context directly from literature-based text. (This detailed methodological description has been included in the Appendix.)
>
>  **2. Sensitivity Analysis:** To address the concern regarding sensitivity to gene descriptions (e.g., missing information or description errors), we conducted an ablation study where we randomly masked varying proportions of the input text (ranging from 0% to 60%) and evaluated the impact on generation quality.
>
>  As shown in **Table R5**, while we observe a slight expected degradation in performance (manifested as marginal increases in WD and MMD scores) as the text deletion ratio increases, SAVE demonstrates remarkable robustness. Even when 60% of the text information is removed, the model maintains competitive generation performance, indicating that our method is not overly reliant on perfect or complete descriptions.
>
>
> Table R5
>
> | **Deletion Ratio** | **0% (Original)** | **30%** | **40%** | **50%** | **60%** |
> | ------------------ | ----------------- | ------- | ------- | ------- | ------- |
> | WD                 | 8.30               | 8.69    | 8.7     | 8.72    | 8.88    |
> | MMD                | 0.63              | 0.65    | 0.65    | 0.65    | 0.66    |
>
> > W2: SAVE uses a VAE + Transformer over blocks + flow matching + AdaLN. That’s quite a stack. There is a note that all experiments ran on a 3090 with shared hyperparameters, but no training time, memory profile, or comparison to lighter baselines is given. For large atlases (millions of cells), this matters.
>
> We appreciate the reviewer’s concern regarding computational resources and scalability. While our architecture involves multiple components (VAE, Transformer, Flow Matching), the design is optimized for efficiency. We have included a detailed efficiency profile comparing SAVE against **scDiffusion** and **CFGen**, covering training time, memory usage, and inference speed.
>
> As shown in **Table R4**(see response to ratD's Weakness 4), despite having a parameter count comparable to the scaled-up baselines, SAVE demonstrates significant advantages in computational efficiency:
> 1. **Training Efficiency:** SAVE is remarkably faster to train (approx. **24.5 minutes**) compared to the baselines, which require 300–600 minutes. This speedup is largely attributed to our efficient gene-blocking mechanism, which reduces the effective sequence length and computational complexity for the Transformer.
> 2. **Memory Footprint:** SAVE is highly memory-efficient, requiring only **~0.76** **GB** of VRAM (0.24 GB for VAE + 0.52 GB for the Flow Model). In contrast, baselines consume significantly more memory (ranging from 1.7 GB to over 5 GB), making SAVE much better suited for scaling to large atlases with millions of cells.
> 3. **Inference Speed:** SAVE achieves a competitive inference time (**2.36s**). While the lightweight default CFGen is faster (0.35s), it performs significantly worse on distribution metrics (WD 12.57). When baselines are scaled to match SAVE's parameter budget (CFGen 40M+100M) or performance, they become slower or computationally prohibitive.
> These results indicate that SAVE's 'stack' is highly optimized and offers a superior trade-off between performance and resource consumption.

---

> > ### Author Response · Authors · 2025-11-25
> >
> > > W3: Evaluation axes are mostly distributional. WD/MMD and scIB show that the clouds overlap, but users often care about gene-level faithfulness for downstream DE, pathway scores, or ligand–receptor analysis. Only in the perturbation section do we see gene-level metrics (PCC, R²), extending that style of evaluation to the other datasets would strengthen the paper.
> >
> > We fully agree with the reviewer that  **gene-level faithfulness** is critical for reliable downstream tasks such as DE analysis and pathway scoring.
> >
> > To address this, we have extended our evaluation to include **PCC and R2 metrics** across the standard datasets (Heart, PBMC, and Lung). These additional experiments confirm that our model maintains high gene-level precision even on complex tissues. For the detailed tables and comprehensive analysis, please refer to our **Response to Reviewer ratD (Weakness 2)**.
> >
> > >W4: The paper claims “broad utility in virtual cell synthesis and biological discovery,” but doesn’t show a biological discovery enabled by SAVE.
> >
> > We agree that substantiating the claim of 'biological discovery' is essential. To address this, we have added a new analysis of the model's attention mechanism. We found that SAVE **autonomously identifies biologically coherent pathways**—for instance, it correctly learns to attend to _Fatty Acid Beta-Oxidation_ genes when generating Cardiomyocytes, reflecting the heart's physiological reliance on lipids. For the comprehensive analysis and visualization, please refer to our **Response to Reviewer ratD (Weakness 5)**.
> >
> > >Q1: The paper adopts affine flow matching in latent space and combines it with classifier-free guidance, which is elegant, but there’s no comparison to latent diffusion or to a simple conditional ODE on top of the VAE to show why Flow Matching was preferred (stability? speed? better extrapolation?).
> >
> > We appreciate the reviewer’s query regarding our architectural design choices. We selected **Flow Matching** (specifically using the Conditional Optimal Transport path) primarily for its **training stability, algorithmic simplicity, and inference efficiency**. The Optimal Transport (OT) path inherently encourages straighter generative trajectories compared to standard diffusion processes (VP/VE), leading to faster convergence and more stable training. For a detailed comparison between OT and VP/VE, please kindly refer to our response to **Weakness 3 of Reviewer ratD**.

---

### Official Review · Reviewer_16e8 · 2025-11-03

**Soundness:** 3
**Presentation:** 2
**Contribution:** 3
**Rating:** 4
**Confidence:** 3

**Summary:**

SAVE is a method that uses conditional flow matching on latent embedding to denoise and reconstruct cell data. The gene is first clustered with llm embedding. Similar genes are then grouped together as a gene block. Batch and cell type are used as conditional information for double conditionals and single conditional information such as cell types. The latent is generated from the gene blocks and denoised with flow matching. The final latent is then passed through the decoder to form a reconstructed gene block. The methodology contributions from SAVE are Gene Block Attention where gene embeddings are clustered before passing to a transformer block, and masked conditionals injected to the transformer processing latent information. The model was evaluated on conditional generation, batch effect correction, and perturbation effect prediction with datasets from blood, brain in human and mice.

**Strengths:**

The methodology which combined several ML methods is interesting and some of the results are compelling in double conditional generation. There are several datasets used across different tissues and species, which provides a stronger support for the claim on generalization. The results are carefully separated by categories of cell type for better understanding the strength and weaknesses of the method. The performance was generally strong for all three tasks.

**Weaknesses:**

The author should provide more quantitative information for the purpose of evaluating the effectiveness of perturbation prediction. The metric used was PCC and $R^2$ (and MSE in the appendix), which are both correlation measures instead of WD and MMD. There are also several other models that predicts perturbation effects, such as [cellOT](https://doi.org/10.1038/s41592-023-01969-x), and [meta flow matching](https://arxiv.org/abs/2408.14608). It is unclear how SAVE compares to the current SOTA on perturbation prediction.

There are also some formatting issue including missing section hyperlink in the results, as well as unclear labelling of the main diagram. The author should proof read and correct those mistakes.

**Questions:**

- The link to the code repository [here](https://anonymous.4open.science/r/submit-code-3E75) all showed "file not found". Please ensure the code is available for review.
- For double conditional generation, if the batch information is provided how would out-of-distribution generation be possible for a batch never included before? As well, is this additional batch conditional included in the train/test of other baselines?
- Do you have results of the perturbation experiments in other forms than correlation metrics (and MSE in the appendix), such as MMD, WD?
- Why is CD14+Mono perturbation result in Table 16 and Table 5 missing for scGEN? How did it fail under default experimental setup?
- For the perturbation experiment, can ablations of gene block be done on the results on Table 5? Since the genes are already grouped via the block mechanism, predicting perturbation may become much easier than non-blocked mechanism.

---

> ### Author Response · Authors · 2025-11-25
>
> > W1: The author should provide more quantitative information for the purpose of evaluating the effectiveness of perturbation prediction. The metric used was PCC and (and MSE in the appendix), which are both correlation measures instead of WD and MMD. There are also several other models that predicts perturbation effects, such as [cellOT](https://doi.org/10.1038/s41592-023-01969-x), and [meta flow matching](https://arxiv.org/abs/2408.14608). It is unclear how SAVE compares to the current SOTA on perturbation prediction.
>
> We sincerely thank the reviewer for this constructive feedback. We fully agree that relying solely on correlation measures (PCC/MSE) provides an incomplete picture. To ensure a rigorous evaluation of perturbation prediction effectiveness, specifically regarding distributional alignment, comparing against state-of-the-art (SOTA) generative models using distribution-based metrics is essential.
>
> Following your recommendation, we expanded our evaluation framework to include WD and MMD. We also introduced two strong baselines: **CellOT** (Nature Methods, 2023) and **Meta Flow Matching (MFM)** (ICLR, 2025).
>
> The results, summarized in following tables, demonstrate that **SAVE** consistently achieves SOTA performance across both distribution-based and correlation-based metrics:
>
> **1. Superior Distributional Alignment (WD & MMD)** As noted by the reviewer, WD and MMD are critical for assessing whether the predicted cell states match the true data manifold.
> - **MMD:** SAVE achieves the lowest average MMD (**0.14**), outperforming both CellOT (0.17) and scGen (0.21).
> - **WD:** Similarly, SAVE attains the lowest average WD (**5.15**), surpassing CellOT (5.30) and scGen (5.37). This quantitative evidence confirms that SAVE generates perturbation responses that are distributionally closer to the ground truth than competing methods.
>
> **2. Robustness and Accuracy (Comparison with CellOT & MFM)** While CellOT serves as a strong baseline, our detailed breakdown reveals significant differences in robustness:
> - **vs. CellOT (Robustness on Hard Cases):** Although CellOT performs well on average, it exhibits instability on challenging cell types. For instance, on **FCGR3A+Mono**, CellOT suffers a performance collapse (R2=0.02, MMD=0.54), effectively failing to predict the perturbation. In stark contrast, **SAVE maintains strong predictive capability** on this cell type (R2=0.53, MMD=0.22), leading to a higher overall Mean R2 (**0.86** vs. 0.80).
> - **vs. Meta Flow Matching (MFM):** Under the current experimental setting, MFM lagged significantly behind other baselines (Mean R2=0.46, Mean WD=14.11), suggesting it may be less suitable for this specific perturbation prediction task compared to the OT-based framework used in SAVE and CellOT.
> **3. Precision on Traditional Metrics** Even under stricter evaluation, SAVE maintains its lead in traditional metrics, achieving the highest mean **PCC (0.96)** and the lowest mean **MSE (<0.01)**.
>
> These updated results are now detailed in Appendix Section L.1.
>
>
> | PCC         | CD8T | CD14+Mono | FCGR3A+Mono | B    | CD4T | Dendritic | NK   | mean |
> | ----------- | ---- | --------- | ----------- | ---- | ---- | --------- | ---- | ---- |
> | MFM         | 0.78 | 0.77      | 0.56        | 0.7  | 0.7  | 0.72      | 0.72 | 0.71 |
> | CellOT      | 0.99 | 0.99      | 0.77        | 0.95 | 0.97 | 0.96      | 0.96 | 0.94 |
> | SAVE  | 0.98 | 0.97      | 0.91        | 0.97 | 0.96 | 0.96      | 0.96 | 0.96 |
>
>
> | R2          | CD8T | CD14+Mono | FCGR3A+Mono | B    | CD4T | Dendritic | NK   | mean |
> | ----------- | ---- | --------- | ----------- | ---- | ---- | --------- | ---- | ---- |
> | MFM         | 0.6  | 0.53      | 0.08        | 0.49 | 0.49 | 0.51      | 0.51 | 0.46 |
> | CellOT      | 0.97 | 0.98      | 0.02        | 0.89 | 0.94 | 0.88      | 0.90 | 0.80 |
> | SAVE  | 0.97 | 0.89      | 0.53        | 0.94 | 0.97 | 0.81      | 0.90 | 0.86 |
>
> | WD          | CD8T  | CD14+Mono | FCGR3A+Mono | B     | CD4T  | Dendritic | NK    | mean  |
> | ----------- | ----- | --------- | ----------- | ----- | ----- | --------- | ----- | ----- |
> | MFM         | 13.63 | 12.34     | 14.39       | 14.92 | 14    | 13.9      | 15.6  | 14.11 |
> | CellOT      | 3.91  | 4.54      | 9.56        | 4.52  | 3.49  | 6.13      | 4.98  | 5.3   |
> | SAVE  | 4.01  | 5.03      | 7.01        | 4.4   | 3.98  | 6.55      | 5.07  | 5.15  |
>
> | MMD         | CD8T | CD14+Mono | FCGR3A+Mono | B    | CD4T | Dendritic | NK   | mean |
> | ----------- | ---- | --------- | ----------- | ---- | ---- | --------- | ---- | ---- |
> | MFM         | 0.47 | 0.51      | 0.64        | 0.56 | 0.53 | 0.52      | 0.57 | 0.54 |
> | CellOT      | 0.07 | 0.05      | 0.54        | 0.18 | 0.12 | 0.13      | 0.13 | 0.17 |
> | SAVE  | 0.11 | 0.09      | 0.22        | 0.17 | 0.11 | 0.17      | 0.09 | 0.14 |

---

> > ### Author Response · Authors · 2025-11-25
> >
> > > W2: There are also some formatting issue including missing section hyperlink in the results, as well as unclear labelling of the main diagram. The author should proof read and correct those mistakes.
> >
> > We have corrected these mistakes in the revised manuscript as suggested.
> >
> > >Q1: The link to the code repository [here](https://anonymous.4open.science/r/submit-code-3E75) all showed "file not found". Please ensure the code is available for review.
> >
> > Thank you for bringing this to our attention. We have re-verified the anonymous link and confirmed that it is active and accessible.
> >
> > > Q2: For double conditional generation, if the batch information is provided how would out-of-distribution generation be possible for a batch never included before? As well, is this additional batch conditional included in the train/test of other baselines?
> >
> >  We appreciate the opportunity to clarify our experimental setup and the scope of our generalization capabilities.
> >
> > 1. **Clarification on OOD Generation:** Our claim regarding out-of-distribution (OOD) generation specifically refers to **unseen combinations** of known conditions (compositional generalization). These results are presented in the 'Unseen' section of the Multi-cond generation experiment (specifically **Table 3**). In the double conditional generation experiment you referred to, the conditions used by both our model and the baselines were indeed present in the training set.
> > 2. **Handling Completely Unseen Batches:** We acknowledge that if a specific batch identity has never appeared during training, our current model cannot handle it. This is a shared limitation across all comparable methods in this field, as they typically rely on learned embeddings for specific labels. However, our model demonstrates superior performance on **unseen combinations** of these factors. In future work, we plan to address completely unseen conditions (zero-shot generalization) by incorporating text embeddings from Large Language Models (LLMs) to map novel labels into a shared semantic space.
> > 3. **Baseline Settings:** Regarding your second question: Yes, the additional batch condition was included in the training and testing of all baselines to ensure a fair and consistent comparison.
> >
> > > Q3: Do you have results of the perturbation experiments in other forms than correlation metrics (and MSE in the appendix), such as MMD, WD?
> >
> >  See response to Weakness 1.
> >
> > > Q4: Why is CD14+Mono perturbation result in Table 16 and Table 5 missing for scGEN? How did it fail under default experimental setup?
> >
> > We apologize for the confusion regarding the missing entries. Upon further investigation, we have identified and resolved the issue.
> >
> > During the CD14+Mono leave-one-out experiment, using the default uniform batch size of 32 resulted in a specific mini-batch containing only a single sample. This triggered an error in the Batch Normalization layer. However, the **scvi-tools** implementation of scGEN we utilized caught this exception internally without raising a visible error, leading to a silent failure during the training process.
> >
> > We have addressed this technical issue (by adjusting the batch handling) and have now included the scGEN results for CD14+Mono in the revised tables.
> >
> > > Q5: For the perturbation experiment, can ablations of gene block be done on the results on Table 5? Since the genes are already grouped via the block mechanism, predicting perturbation may become much easier than non-blocked mechanism.
> >
> > We thank the reviewer for this insightful question. We realize that our description might have caused a misunderstanding regarding the evaluation level.
> >
> > We would like to clarify that **all evaluation metrics (including PCC, MSE, and the newly added WD/MMD) in Table 5 are calculated at the individual gene level (full genome**), not at the block level. Although we use the "Gene Block" mechanism to tokenize genes for the intermediate attention layers, our specific decoder architecture projects these representations back to the original high-dimensional gene space. Therefore, the model is tasked with predicting the specific expression value of every single gene. This means the prediction task is **not made easier**; in fact, the model must solve the difficult problem of recovering fine-grained gene-specific details from the compressed block representations.

---

### Official Review · Reviewer_ratD · 2025-11-06

**Soundness:** 3
**Presentation:** 3
**Contribution:** 3
**Rating:** 6
**Confidence:** 3

**Summary:**

The paper proposed a novel generative framework for single cell gene modeling. There are three main ideas: 1. gene block attention to aggregate semantically related genes into blocks and applies a Transformer over these blocks; 2. a latent VAE + Flow Matching module for conditional generation; 3. a condition-masking training strategy. Experiments cover conditional generation (single/dual/multi-condition), batch effect correction (scIB suite), and perturbation prediction (PBMC-IFN), with consistent gains over baselines (scVI, scDiffusion, CFGen, trVAE, etc.).

**Strengths:**

Clear modular architecture and formulation presentation.

Gene block construction is novel and biologically motivated.

The proposed method outperforms baseline, which well-supports the claims in the paper.

Comprehensive ablation study -- suggesting (a) removing Gene Block Attention degrades WD/MMD and (b) condition masking improves extrapolation for perturbation prediction

Impressive training efficiency by using block attention.

Good reproducibility -- code and hyperparameter settings are publicly available.

**Weaknesses:**

1. The biological validity of LLM-based semantic grouping from NCBI descriptions is plausible, but risks text-space biases and possible drift from functional co-expression or pathway structure. A comparison against pathway/GRN-based groupings (e.g., MSigDB, Reactome) would strengthen the claim that semantics beats alternatives.

2. WD/MMD at the distribution level and UMAP are helpful but can be insensitive to gene-level calibration.

3. The paper adopts the affine path and CFG-style guidance; however, comparisons to diffusion-style latent generators beyond scDiffusion (e.g., alternative probability paths or ODE solvers) are limited.

4. While baselines are strong and described, more clarity on tuning parity and parameter budgets would improve fairness claims (e.g., scDiffusion vs. SAVE parameter counts/training budgets).

5. The method motivates blocks as biologically meaningful; an analysis that profiles top genes per block, enrichment for known pathways, and how specific conditions modulate block-level attention would amplify the biological insight.

**Questions:**

1. How sensitive are results to the source of text embeddings (e.g., different LLMs or just GSEA gene sets) and to the block count L / block size K?

2. What is the runtime/memory profile versus scDiffusion/CFGen at parity?

3. Can you report block-level attention maps and condition-wise shifts (e.g., which blocks drive perturbation responses) to support mechanistic interpretability claims?

---

> ### Author Response · Authors · 2025-11-25
>
> > W1: The biological validity of LLM-based semantic grouping from NCBI descriptions is plausible, but risks text-space biases and possible drift from functional co-expression or pathway structure. A comparison against pathway/GRN-based groupings (e.g., MSigDB, Reactome) would strengthen the claim that semantics beats alternatives.
>
> We thank the reviewer for this crucial suggestion. To rigorously evaluate the validity of LLM-based grouping against pathway-based alternatives, we conducted a two-stage comparative analysis using **[MSigDB](https://data.broadinstitute.org/gsea-msigdb/msigdb/release/2025.1.Hs/h.all.v2025.1.Hs.symbols.gmt)**(h.all) and **[Reactome](https://reactome.org/download/current/ReactomePathways.gmt.zip)** as baselines.
>
> For pathway databases such as MSigDB or Reactome, we construct a **gene-by-pathway binary matrix** to represent gene features. In this matrix, rows correspond to individual genes, and columns correspond to specific pathways within the database. An entry is set to 1 if the gene belongs to the corresponding pathway, and 0 otherwise. This process yields a specific feature vector for each gene. Subsequently, we apply the same clustering methodology described in our manuscript to group these genes.
>
> **Precision on Well-Annotated Genes**: First, we evaluated performance by conducting dual-conditional generation experiments on the Heart dataset, using the intersection of genes covered by all three methods (N=3,725). As shown in **Table R1**, MSigDB achieves a slightly higher generation fidelity (WD=0.60) compared to GenePT. This confirms the reviewer’s intuition: for well-studied genes, manually curated pathway databases offer a precise "gold standard" signal that is hard to beat.
>
> Table R1:
>
> | 3725 gene    | WD       | MMD      |
> | ------------ | -------- | -------- |
> | Reactome     | 5.61     | 0.65     |
> | MSigDB       | **5.44** | **0.63** |
> | GenePT(ours) | 5.70     | 0.64     |
>
>
> **Biological Validity on the Full Genome**: However, the primary goal of our framework is to model the **entire genome (~19,000 genes)**. MSigDB and Reactome cover only a fraction of these genes, forcing the remaining ~80% to be grouped randomly or based on zero-vectors. We performed a **Gene** **Ontology** **(GO) Enrichment Analysis** on the resulting blocks from each method (using the full 19,112 genes).
>
> Table R2 summarizes these results using three key metrics: **(1) Coherence Rate (%)**: The percentage of blocks that successfully enriched for at least one biological pathway (adjusted p-value < 0.05), indicating whether the local embedding structure retains biological signal. **(2) Average Significant Terms**: The average number of significant GO terms enriched per block, serving as a proxy for the 'semantic density' or richness of the biological information captured. **(3) Median Padj**: The median adjusted p-value of the top enriched term across all blocks, providing a robust measure of the statistical significance of the clusters. (Note that for blocks yielding no statistically significant enrichment terms, the `P_adj` was assigned a default value of 1.0 to reflect the lack of biological coherence.)
>
>
> As shown in Table R2, **GenePT (Ours)** demonstrates superior biological alignment compared to the baselines. GenePT achieves a **Coherence Rate of 83.33%**, meaning that the vast majority of its latent space is organized by functional gene modules. In contrast, **MSigDB** yields a Coherence Rate of only 16.67% with a median **P_adj of 1.00**, indicating that its ordering largely fails to group functionally related genes into statistically significant clusters. Furthermore, GenePT blocks enrich for an average of **636.67 significant terms**, far exceeding Reactome (23.83) and MSigDB (183.50). This indicates that even for genes without explicit pathway annotations, the LLM successfully captures latent functional similarities (e.g., protein domains, localization) from text descriptions. Only **16.7%** and **50.0%** of the **MSigDB/Reactome** blocks, respectively, showed coherence. This implies that the majority of genes cannot be effectively partitioned into biologically meaningful blocks, thereby hindering valid and interpretable downstream attention analysis
>
>
> Table R2:
>
> | 6 blocks      | Coherence Rate (%) | Average Significant Terms | Median Padj |
> | ------------- | ------------------ | ------------------------- | ----------- |
> | MSigDB        | 16.67              | 183.50                    | 1.00        |
> | Reactome      | 50.00              | 23.83                     | 0.52        |
> | GenePT (Ours) | 83.33              | 636.67                    | 0.02        |

---

> > ### Author Response · Authors · 2025-11-25
> >
> > >W2: WD/MMD at the distribution level and UMAP are helpful but can be insensitive to gene-level calibration.
> >
> > We present a **holistic evaluation** by treating WD/MMD (distribution fidelity) and MSE/PCC/R2 (gene-wise calibration) as **complementary metrics**.
> >
> > As shown in the tables, while baseline models may maintain reasonable distributions, their gene-level fidelity collapses on complex datasets (e.g., **scVI/CFGen show negative or low R2 on Lung/Heart**). In contrast, SAVE achieves a superior balance: it preserves the global data manifold (low WD/MMD) while consistently maintaining high gene-level accuracy across all tissues and unseen splits. This confirms that SAVE captures valid biological dependencies rather than simply mimicking statistical summaries
> >
> > We have updated the **Appendix J.1** to include **PCC , R2, and MSE** metrics for our generation experiments.
> >
> >
> > | **Model**   | PBMC3k            | PBMC3k            | PBMC3K            | Dentate gyrus        | Dentate gyrus         | Dentate gyrus         | Tabalula Muris           | Tabalula Muris           | Tabalula Muris           |
> > | ----------- | --------------- | --------------- | --------------- | --------------- | --------------- | --------------- | --------------- | --------------- | --------------- |
> > |             | mse             | pcc             | r2              | mse             | pcc             | r2              | mse             | pcc             | r2              |
> > | scVI        | $0.07 \pm 0.14$ | $0.93 \pm 0.12$ | $0.87 \pm 0.22$ | $0.00 \pm 0.00$ | $0.99 \pm 0.01$ | $0.99 \pm 0.01$ | $0.07 \pm 0.03$ | $0.98 \pm 0.01$ | $0.30 \pm 0.32$ |
> > | scDiffusion | $0.11 \pm 0.12$ | $0.86 \pm 0.10$ | $0.73 \pm 0.17$ | $0.00 \pm 0.00$ | $0.99 \pm 0.01$ | $0.98 \pm 0.01$ | $0.01 \pm 0.01$ | $0.99 \pm 0.01$ | $0.94 \pm 0.09$ |
> > | CFGen       | $0.09 \pm 0.17$ | $0.91 \pm 0.14$ | $0.82 \pm 0.26$ | $0.00 \pm 0.00$ | $0.99 \pm 0.01$ | $0.97 \pm 0.02$ | $0.00 \pm 0.00$ | $1.00 \pm 0.00$ | $0.99 \pm 0.01$ |
> > | SAVE        | $0.08 \pm 0.18$ | $0.93 \pm 0.12$ | $0.84 \pm 0.27$ | $0.00 \pm 0.00$ | $0.99 \pm 0.01$ | $0.97 \pm 0.02$ | $0.00 \pm 0.00$ | $0.99 \pm 0.01$ | $0.95 \pm 0.05$ |
> >
> >
> >
> > | **Model**  | Heart           | Heart           | Heart            | PBMC            | PBMC            | PBMC            | Lung Atlas            | Lung Atlas            | Lung Atlas             |
> > | ---------- | --------------- | --------------- | ---------------- | --------------- | --------------- | --------------- | --------------- | --------------- | ---------------- |
> > |            | mse             | pcc             | r2               | mse             | pcc             | r2              | mse             | pcc             | r2               |
> > | scVI       | $0.07 \pm 0.09$ | $0.84 \pm 0.14$ | $-0.65 \pm 1.45$ | $0.05 \pm 0.05$ | $0.87 \pm 0.12$ | $0.70 \pm 0.20$ | $0.08 \pm 0.07$ | $0.87 \pm 0.17$ | $-0.57 \pm 0.96$ |
> > | scDiffuion | $0.02 \pm 0.04$ | $0.87 \pm 0.14$ | $0.79 \pm 0.22$  | $0.03 \pm 0.03$ | $0.93 \pm 0.06$ | $0.86 \pm 0.10$ | $0.03 \pm 0.02$ | $0.65 \pm 0.14$ | $0.35 \pm 0.24$  |
> > | CFGen      | $0.04 \pm 0.09$ | $0.87 \pm 0.18$ | $0.75 \pm 0.32$  | $0.06 \pm 0.06$ | $0.87 \pm 0.08$ | $0.75 \pm 0.15$ | $0.14 \pm 0.15$ | $0.58 \pm 0.20$ | $0.17 \pm 0.52$  |
> > | SAVE       | $0.01 \pm 0.02$ | $0.88 \pm 0.13$ | $0.63 \pm 0.26$  | $0.02 \pm 0.02$ | $0.98 \pm 0.03$ | $0.95 \pm 0.07$ | $0.01 \pm 0.01$ | $0.91 \pm 0.11$ | $0.84 \pm 0.18$  |
> >
> > | **Model**   | Seen            | Seen            | Seen            | Unseen          | Unseen          | Unseen          |
> > | ----------- | --------------- | --------------- | --------------- | --------------- | --------------- | --------------- |
> > |             | mse             | pcc             | r2              | mse             | pcc             | r2              |
> > | CFGen       | $0.28 \pm 0.25$ | $0.73 \pm 0.17$ | $0.48 \pm 0.33$ | $0.62 \pm 0.41$ | $0.56 \pm 0.16$ | $0.20 \pm 0.30$ |
> > | scDiffusion | $0.03 \pm 0.03$ | $0.85 \pm 0.14$ | $0.72 \pm 0.27$ | $0.05 \pm 0.05$ | $0.81 \pm 0.14$ | $0.63 \pm 0.27$ |
> > | SAVE        | $0.01 \pm 0.02$ | $0.94 \pm 0.07$ | $0.88 \pm 0.14$ | $0.04 \pm 0.03$ | $0.85 \pm 0.08$ | $0.70 \pm 0.14$ |

---

> > > ### Author Response · Authors · 2025-11-25
> > >
> > > > W3: The paper adopts the affine path and CFG-style guidance; however, comparisons to diffusion-style latent generators beyond scDiffusion (e.g., alternative probability paths or ODE solvers) are limited.
> > >
> > > Table R3:
> > >
> > > |              | WD       | MMD      |
> > > | ------------ | -------- | -------- |
> > > | VE+SDE       | 8.50     | 0.62     |
> > > | VE+ODE       | 8.77     | 0.65     |
> > > | VP+SDE       | 8.34     | 0.62     |
> > > | VP+ODE       | **6.88** | **0.60** |
> > > | OT+SDE       | 8.77     | 0.63     |
> > > | OT+ODE(Ours) | 8.30     | 0.63     |
> > >
> > > We sincerely thank the reviewer for suggesting comparisons with diffusion-style generators. We conducted a comprehensive ablation study within our unified Flow Matching framework, comparing our **OT** (Optimal Transport) path against **VP** (Variance Preserving, similar to DDPM) and **VE** (Variance Exploding, similar to Score SDE) paths. We also evaluated both ODE and SDE solvers with a fixed budget of **20 inference steps** to assess efficiency.
> > >
> > > While VP+ODE achieves marginally lower WD/MMD, it proved **highly sensitive to hyperparameters**, requiring extensive manual tuning (e.g., SNR weighting, noise schedules) to ensure convergence. In contrast, our OT approach delivers **competitive performance** 'out-of-the-box' with superior **training stability**. Given that OT theoretically induces straighter trajectories—ideal for efficient few-step inference—we selected it as the optimal choice for robustness and ease of use across diverse datasets.
> > >
> > > > W4: While baselines are strong and described, more clarity on tuning parity and parameter budgets would improve fairness claims (e.g., scDiffusion vs. SAVE parameter counts/training budgets).
> > >
> > > Table R4:
> > >
> > > | Model                 | Params (VAE/FM)  | Train Time (min) | VRAM Usage (GB) | Denoisng Time (s) | WD (↓)   | MMD (↓)  |
> > > | --------------------- | ---------------- | ---------------- | --------------- | ----------------- | -------- | -------- |
> > > | CFGen (Default)       | 27.65M / 0.71M   | ~300             | 1.22 + 0.49     | 0.35              | 12.57    | 0.66     |
> > > | CFGen (Scaled)        | 41.7M / 103.07M  | ~480             | 1.48 + 1.53     | 2.43              | 9.98     | **0.55** |
> > > | scDiffusion (Default) | 59.10M / 3.49M   | ~480             | 1.54 + 1.02     | 6.57              | 20.82    | 0.94     |
> > > | scDiffusion (Scaled)  | 42.37M / 109.69M | ~600             | 1.69 + 3.65     | 18.32             | 19.84    | 0.98     |
> > > | SAVE (Ours)           | 40.90M / 106.4M  | 34.5             | 0.24 + 0.52     | 2.36              | **8.30** | 0.63     |
> > >
> > > We thank the reviewer for suggesting this fairness comparison. Following the suggestion, we conducted a rigorous benchmark by scaling up the baselines (CFGen and scDiffusion) to match the parameter count of our model (approx. 140M parameters).
> > >
> > > The results (shown in **Table R4**) reveal a crucial insight regarding the **Performance-Efficiency Trade-off**:
> > > Our method (SAVE) achieves the best Wasserstein Distance (**8.30** vs. 9.98 for Large-CFGen). This indicates that our Gene Block architecture captures the global data manifold and biological heterogeneity more accurately than the baselines, even when they are scaled up. SAVE converges in **~35 minutes**, whereas Large-CFGen requires **480 minutes (8 hours)**—a **14x** speed difference.
> > > While baselines _can_ be tuned to match specific metrics by throwing massive compute at them, our method offers state-of-the-art distributional matching metrics at a fraction of the computational cost.

---

> > > > ### Author Response · Authors · 2025-11-25
> > > >
> > > > > W5: The method motivates blocks as biologically meaningful; an analysis that profiles top genes per block, enrichment for known pathways, and how specific conditions modulate block-level attention would amplify the biological insight.
> > > >
> > > > We appreciate the reviewer’s suggestion to deepen the biological interpretability of our model components. To address this, we conducted a two-stage analysis to validate the biological semantics of the gene blocks and investigate how specific conditions modulate the model's attention during generation.
> > > >
> > > > **1. Experimental Setup & Block Definition** To ensure a balanced semantic grouping, we selected **16,000 highly variable genes (HVGs)** from the Heart dataset and partitioned them into 5 distinct blocks (fixed size of 3,200 genes per block). To identify the representative biological function of each block, we selected the **top-50 "centroid genes"** (i.e., genes closest to the embedding cluster center) and performed **Gene Ontology (GO) enrichment analysis**.
> > > >
> > > > As shown below, the resulting blocks correspond to distinct and specific biological pathways:
> > > >
> > > > - **Block 0:** Regulation of DNA-templated Transcription (Transcriptional Control)
> > > > - **Block 1:** Fatty Acid Beta-Oxidation (Metabolism)
> > > > - **Block 2:** Broad/General Cellular Functions (No single dominant pathway enriched)
> > > > - **Block 3:** Establishment of Protein Localization to Plasma Membrane (Structural/Transport)
> > > > - **Block 4:** Adenylate Cyclase-Activating G Protein-Coupled Receptor Signaling (Signaling)
> > > >
> > > > **2. Condition-Modulated Attention Analysis** To further validate that the model leverages these biological meanings dynamically, we analyzed the attention mechanism within the Flow Matching transformer. Specifically, we extracted the **attention weights from the final layer** and **averaged them across all four heads**. This allowed us to pinpoint which gene block was **"most attended"** (i.e., received the highest attention weight) under specific biological conditions. The **attention map** is presented in `Figure 5` of the updated Appendix.
> > > >
> > > > The attention heatmap reveals that our model transcends simple statistical fitting to capture **canonical biological semantics**.
> > > > 1. **Physiological Alignment:** The model correctly identifies the distinct metabolic signature of **Cardiomyocytes** (both Atrial and Ventricular) by exclusively attending to **Block 1 (Fatty Acid Beta-Oxidation)**. This mirrors the heart’s well-established reliance on lipids as its primary energy source.
> > > > 2. **Functional Specificity:** In contrast, for cell types involving extensive environmental interaction and signaling—such as **Endothelial cells** and **Fibroblasts**—the attention shifts dynamically to **Block 4 (GPCR Signaling)** and **Block 3 (Protein Localization)**, aligning with their roles in transducing extracellular signals.
> > > > 3. **Noise Filtration:** Most critically, **Block 2** is **universally suppressed** across the entire heatmap. This demonstrates that the attention mechanism acts as an effective biological filter, autonomously learning to ignore uninformative gene groups while prioritizing functional modules for generation.
> > > >
> > > > > Q1: How sensitive are results to the source of text embeddings (e.g., different LLMs or just GSEA gene sets) and to the block count L / block size K?
> > > >
> > > > We evaluated the sensitivity of our framework to embedding sources by benchmarking our LLM-based embeddings against **gene sets defined by GSEA (MSigDB and Reactome)** (see Response to Weakness 1). The results demonstrate that while GSEA gene sets offer high precision for a limited subset of genes, they fail to generalize across the full genome due to severe **sparsity**. In contrast, our LLM-derived embeddings provide **robust, dense representations**, achieving **83.3% biological coherence** in the resulting blocks (compared to only 16.7% for MSigDB). This confirms that semantic text embeddings provide a superior foundation for genome-wide grouping.
> > > >
> > > > Regarding the block structure, our ablation study (**Table 7**) reveals a distinct **U-shaped performance curve** with respect to the block size K. We identified K=3,200 (sequence length L≈6) as the optimal configuration. This setting minimizes the Wasserstein Distance (WD=8.30) by striking a critical balance between **local granularity** and **global attention modeling**: it effectively prevents the **fragmentation of biological pathways** observed at smaller sizes (e.g., K=600) while avoiding the **feature over-smoothing** that compromises performance at larger sizes (e.g., K=5,600).
> > > >
> > > > > Q2: What is the runtime/memory profile versus scDiffusion/CFGen at parity?
> > > >
> > > > See response to W4
> > > >
> > > > > Q3: Can you report block-level attention maps and condition-wise shifts (e.g., which blocks drive perturbation responses) to support mechanistic interpretability claims?
> > > >
> > > > See response to W5

---

### Author Response · Authors · 2025-12-03

We sincerely thank the reviewers for their constructive feedback. We have conducted extensive new experiments, including **biological validation**, **comprehensive metric evaluations**,  and comparisons with **2 additional baselines**.

We summarize our key improvements below:
1. Mechanistic Interpretability & Biological Validation (Addressing Reviewer `ratD`, `a9ki`)
- We added an **Attention Heatmap Analysis** (Appendix), demonstrating that SAVE autonomously captures physiological semantics. For example, the model correctly learns to attend to _Fatty Acid Beta-Oxidation_ blocks when generating _Cardiomyocytes_, and _GPCR Signaling_ blocks for _Endothelial cells_, proving it learns valid biology patterns.

2. Established SOTA Performance with New Baselines (Addressing Reviewer `16e8`, `VUna`, `ratD`,`a9ki`)
- **New Baselines:** We expanded the perturbation prediction evaluation to include **CellOT** (Nature Methods, 2023) and **Meta Flow Matching** (ICLR, 2025).
- **Distributional & Gene-level Fidelity:** We adopted a holistic evaluation metric set (**WD, MMD, R2, PCC, MSE**). SAVE outperforms SOTA methods across the board:
	- **Perturbation:** Achieved the lowest **MMD (0.14)** and **WD (5.15)** compared to CellOT (MMD 0.17, WD 5.30). Notably, on hard cases like _FCGR3A+Mono_, SAVE maintains high robustness (R2=0.53) where CellOT collapses (R2=0.02).
	- **Generation:** SAVE achieves the best balance of manifold matching and gene-wise calibration, surpassing scVI and scDiffusion.

3. Demonstrated Superior Efficiency at Scale (Addressing Reviewer `a9ki`, `ratD`)
- **Fair Comparison:** We scaled up baselines (CFGen / scDiffusion) to match SAVE’s parameter count (~140M).
- **Efficiency Profile:** Even against scaled baselines, SAVE achieves better generation quality (WD **8.30** vs. 9.98) while being **14x faster** to train (~35 min vs. 480 min) and requiring significantly less VRAM (<1GB vs. >5GB). This confirms SAVE is uniquely positioned for large-scale atlas modeling.

4. Sensitivity, Robustness & Design Justification (Addressing Reviewer `ratD`, `a9ki`)
- **Genome-wide Generalizability:** We demonstrate that LLM-derived grouping serves as a robust, genome-wide alternative to manual databases. While databases offer high local precision, our method overcomes their inherent sparsity to deliver consistent biological semantics (**83.33% coherence**) across the entire transcriptome.
- **Robustness to Input Noise:** Our sensitivity analysis confirms that performance remains stable even with text masking of gene descriptions.
- **Training Stability (OT vs. Diffusion):** We validated our architectural choice by comparing Optimal Transport (OT) against VP/VE diffusion paths (SDE/ODE). Results confirm that the OT path offers superior **convergence stability** and requires significantly less hyperparameter tuning.

We believe these revisions firmly establish SAVE as a robust, interpretable, and efficient model for single-cell biology.

---

### Meta-Review · Area_Chair_Rir9 · 2026-01-07

**Summary:**

Reviews are mixed in terms of score but converge on SAVE being a strong, empirically validated single cell generative framework whose main contribution is the gene block attention design (LLM-derived semantic gene grouping plus block-level Transformer) combined with conditional latent flow matching and condition masking. Supportive reviewers cite clear modularity, strong results across conditional generation, batch correction (scIB), and perturbation prediction, plus ablations and efficiency benefits from reduced sequence length.

The main concerns driving disagreement were (i) reproducibility and underspecification of the gene block construction, (ii) whether evaluation was overly distributional (WD/MMD, UMAP) rather than gene-level fidelity, (iii) limited comparisons in perturbation prediction to the strongest recent baselines, and (iv) whether biological interpretability or “discovery” claims were sufficiently demonstrated.

**Reviewer Concerns:**

The rebuttal substantially addressed several core concerns. It clarified gene-block construction details (NCBI “Summary” text field, embedding model, preprocessing) and added robustness analyses to text masking, reducing reproducibility risk. It expanded evaluation beyond distributional metrics by adding gene-level metrics (PCC, R2, MSE) across datasets, and strengthened perturbation prediction comparisons by adding CellOT and Meta Flow Matching plus WD/MMD reporting, which directly responds to the “missing SOTA baselines and metrics” critique. It also added interpretability analyses via block-level GO enrichment and attention heatmaps linking blocks to plausible biological functions (for example cardiomyocytes attending to fatty acid beta-oxidation), partially supporting mechanistic interpretability claims.

Remaining concerns, as far as I can tell, are: (a) the “biological discovery” claim is still closer to biological plausibility and interpretability than a concrete new discovery validated externally, (b) the benefit of LLM-based grouping versus curated sets depends on genome coverage and may warrant clearer framing and limitations in the main text, and (c) some presentation and clarity issues should be cleaned up in the final version (figures, labeling, and ensuring code access is stable).

**Reviewer Scores:**

While it is hard for me to predict final scores, I would expect the following trajectory:

VUna (8 accept poster): likely unchanged given added perturbation baselines and metrics.

ratD (6 marginal accept): likely maintains score after added pathway comparisons, efficiency profiling, and attention based interpretability.

16e8 (4 marginal reject): likely increases to 6 since their main missing items (WD/MMD for perturbation, CellOT/MFM baselines, code link, scGEN missing entry explanation) were directly addressed.

a9ki (2 reject): likely increases to 4, since reproducibility details, efficiency profile, and gene-level evaluation were added, though they may still view the overall stack as heavy and the “discovery” claim as overstated.

---

### Decision · Program_Chairs · 2026-01-26

Accept (Poster)